# Reduced-Order Neural Operators: Learning Lagrangian Dynamics on Highly Sparse Graphs

## Abstract

We propose accelerating the simulation of Lagrangian dynamics, such as fluid flows, granular flows, and elastoplasticity, with neural-operator-based reduced-order modeling. While full-order approaches simulate the physics of every particle within the system, incurring high computation time for dense inputs, we propose to simulate the physics on sparse graphs constructed by sampling from the spatially discretized system. Our discretization-invariant reduced-order framework trains on any spatial discretizations and computes temporal dynamics on any sparse sampling of these discretizations through neural operators. Our proposed approach is termed Graph Informed Optimized Reduced-Order Modeling or *GIOROM*. Through reduced order modeling, we ensure lower computation time by sparsifying the system by $6.6$-$32.0\times$, while ensuring high-fidelity full-order inference via neural fields. We show that our model generalizes to a range of initial conditions, resolutions, and materials.

## 1 Introduction

Simulating the dynamics of physical systems is crucial in fields like computational fluid mechanics, digital twins, graphics, and robotics. These spatio-temporal dynamics are often described by partial differential equations (PDEs) in the following form:

$$\mathcal{J}(\boldsymbol{\phi}, \boldsymbol{\nabla}\boldsymbol{\phi}, \boldsymbol{\nabla}^2\boldsymbol{\phi}, \ldots, \dot{\boldsymbol{\phi}}, \ddot{\boldsymbol{\phi}}, \ldots) = \mathbf{0}, \quad \boldsymbol{\phi}(\boldsymbol{X}, t) : \Omega \times \mathcal{T} \to \mathbb{R}^d, \tag{1}$$

where $\boldsymbol{\phi}$ represents a multidimensional vector field that depends on both space and time. The symbols $\boldsymbol{\nabla}$ and $\dot{(\cdot)}$ signify the spatial gradient and time derivative, respectively. Here, $\Omega \subset \mathbb{R}^d$ and $\mathcal{T} \subset \mathbb{R}$ denote the spatial and temporal domains, respectively. In this work, we focus on the deformation map arising from continuum mechanics (Gonzalez & Stuart, 2008), i.e., the position field $\boldsymbol{x} = \boldsymbol{\phi}(\boldsymbol{X}, t)$. Its first and second temporal derivatives denote "velocity" $\boldsymbol{v} = \dot{\boldsymbol{\phi}}$ and "acceleration" $\boldsymbol{a} = \ddot{\boldsymbol{\phi}}$ respectively.

To computationally solve 1, the system is discretized both temporally and spatially. Temporal discretization ($\{t_n\}_{n=0}^N$) breaks down the system's continuous evolution into discrete time steps. After introducing a temporal discretization $\{t_n\}_{n=0}^N$, we solve for a sequence of spatial functions $\{\boldsymbol{\phi}_{t_n}(\boldsymbol{X})\}_{n=0}^N$, where $\boldsymbol{\phi}_{t_n}(\boldsymbol{X}) = \boldsymbol{\phi}(\boldsymbol{X}, t_n)$. Similarly, spatial discretization partitions the physical domain into a discrete set of spatial points, denoted as $\{\boldsymbol{X}^j\}_{j=1}^P$, representing the $P$-point discretization.

Deep-learning-based methods have emerged as an efficient tool for solving these spatio-temporal systems (Azizzadenesheli et al., 2024; Zhang et al., 2023; Cuomo et al., 2022). Instead of solving the PDE explicitly at every time-step, these deep-learning methods implicitly time-step the system via neural network evaluations. Graph Neural Network-based models, such as GNS (Sanchez-Gonzalez et al., 2020), have shown promise in simulating a diverse range of physical systems. A salient feature of GNS is that after training on particular spatial discretizations of the system, it can generalize to other discretizations. However, scaling GNS to large-scale systems can be challenging due to the message-passing operations, which aggregate information between all the neighbors. Furthermore, using a large number of message passing layers leads to over-smoothing. Neural operators Li et al. (2020a); Lu et al. (2021); Kovachki et al. (2023), which learn the mappings between infinite-dimensional function spaces, offer a principled way of dealing with input and output data at arbitrary resolution. In particular, graph-based neural operators Li et al. (2020b;c; 2024) can be applied to

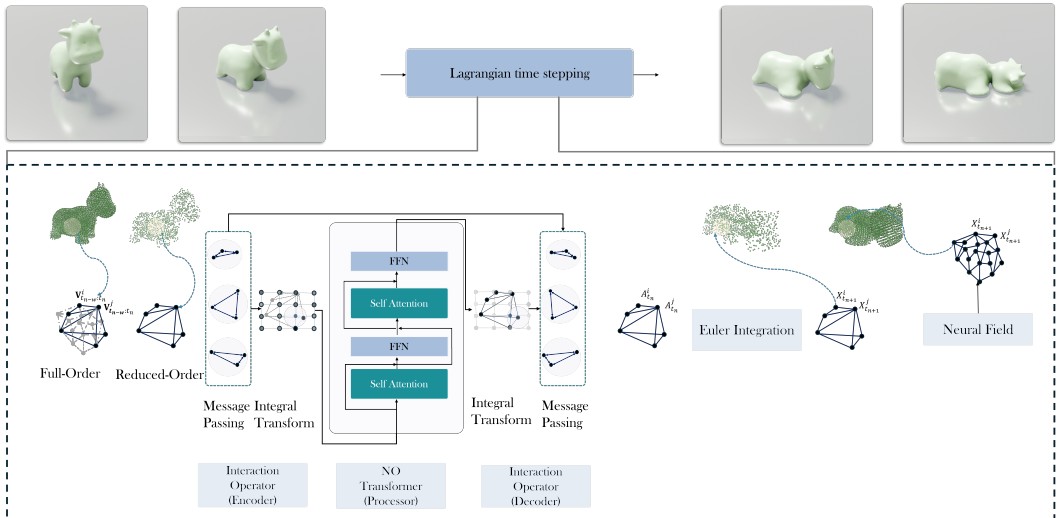

Figure 1: **The overall architecture of GIOROM.** The neural operator $\mathcal{G}_\theta$ predicts the acceleration of a Lagrangian system $\mathbf{A}_{t_k}$ at time $t_k$ from the past $w$ velocity instances $\mathbf{V}_{t_{k-w:k}}$. The positions are derived through Euler integration. The neural field is used to efficiently evaluate the deformation field at arbitrary locations.

graphs of any resolution by taking special care of the construction and aggregation of neighborhoods. However, these methods still have to operate on a very high number of spatial points to ensure high fidelity, just as their classic numerical counterparts. As such, these deep learning models still incur large computation times when operating on highly dense full-order systems (e.g., point clouds).

Reduced-order methods aim to address this computational challenge by operating on reduced-order systems (e.g., a subset of the full-order point clouds) (Benner et al., 2015). In particular, continuous reduced-order modeling approaches (Pan et al., 2023; Chen et al., 2023) create a reduced-order representation for the continuous PDE themselves, not their discretizations. As such, these methods generalize across various spatial discretizations of the system. However, these discretization-agnostic ROM methods are intrusive, in the sense that they require exact PDE information due to the need to solve the PDEs explicitly at every time step, thereby preventing them from being applied to problems where the PDEs are unknown (Lusch et al., 2018).

While, reduced-order methods such as CROM Chen et al. (2023) address computational costs by operating in reduced-order spaces, their flexibility and speed for solving spatio-temporal dynamics is still limited by their reliance on classical numerical PDE solvers. To overcome these limitations, we use the data-driven capabilities of neural operators, which have shown significant accelerations for solving PDEs on different geometries. Specifically, we leverage the graph neural operator as in Li et al. (2024) to handle irregular point clouds. However, the latter model is designed to learn Eulerian formulations of computational fluid dynamics problems without temporal dynamics. Lagrangian dynamics, on the other hand, is influenced by inter-particle interactions, such as collisions. Thus, we propose a discretization-agnostic *Interaction Operator*, which allows for local interactions within the graph neural operator (see Section 4). Moreover, we propose a transformer neural operator in latent space to efficiently model global interactions, see also Figure 1. In summary, we introduce a novel method for learning Lagragian dynamics that reduces the cost for both spatial (via neural fields) and temporal (via neural operators) modeling in a fully discretization-agnostic manner.

Our key contributions can be summarized as follows

- **A framework to learn Lagrangian dynamics on highly sparse graphs:** We propose a spatial-sampling-based reduced order modeling strategy that can accurately and efficiently learn temporal dynamics on very sparse graphs, achieving 6.6-32$\times$ reduction in input size over full-order neural physics solvers, while also delivering high fidelity performance on diverse systems.

- **Graph-based neural operator:** To learn the temporal dynamics, we present a graph-based neural operator transformer that is discretization agnostic.We refer to this model as the time-stepper.
- **Arbitrary spatial evaluation using neural fields:** We leverage continuous reduced-order modeling techniques to evaluate the full-order system at arbitrary spatial points through neural fields.

## 2 RELATED WORKS

**Neural physics solvers**   Neural network-based solvers have shown great success in accelerating physical simulations, including problems in fluid dynamics Sanchez-Gonzalez et al. (2020); Kochkov et al. (2021); Vinuesa & Brunton (2021); Mao et al. (2020); Shukla et al. (2024); Hao et al. (2024), solid mechanics Geist & Trimpe (2021); Capuano & Rimoli (2019); Jin et al. (2023), climate modeling Pathak et al. (2022), and robotics Ni & Qureshi (2022); Kaczmarski et al. (2023). Such approaches can be purely data-driven or physics-informed, i.e., leveraging an underlying PDE Raissi et al. (2019); Sirignano & Spiliopoulos (2018); Richter & Berner (2022); Nam et al. (2024). Moreover, different architectures have been proposed for the neural networks. Approaches based on convolutional neural networks can be used to numerically solve systems of PDE on fixed regular grids Lee & Carlberg (2020a); Maulik et al. (2021); Stoffel et al. (2020); Bamer et al. (2021). For more general meshes, graph neural networks (GNNs) have been proposed, e.g., in the context of mesh-based physics Cao et al. (2022); Pfaff et al. (2020); Han et al. (2022); Fortunato et al. (2022), Lagrangian dynamics Sanchez-Gonzalez et al. (2020), parametric PDEs Pichi et al. (2024), and rigid body physics Kneifl et al. (2024). GNNs can efficiently capture spatial interactions between particles. However, their time complexity scales with the size of the graph since they require message-passing operations on every node. For finer resolutions, this can be computationally prohibitive. Moreover, in their standard formulation, they do not generalize to graphs that have significantly different sizes than the ones seen during training.

**Neural operator models**   Neural operators are a class of discretization agnostic neural network architectures that can generalize to arbitrary discretization of input data. These architectures have been used in solving parametric PDEs Lu et al. (2021); Li et al. (2020c; 2023); Azizzadenesheli et al. (2024); Rahman et al. (2024); Liu-Schiaffini et al. (2024); Kovachki et al. (2023); Rahman et al. (2022a); Liu et al. (2022); Viswanath et al. (2023); Shih et al. (2024); Goswami et al. (2023), fluid dynamics Di Leoni et al. (2023); Wang et al. (2024); Peyvan et al. (2024), protein interactions Liu et al. (2024b;a); Dharuman et al. (2023), 3D physics Xu et al. (2024); White et al. (2023); Bonev et al. (2023); Rahman et al. (2022b); He et al. (2024); Rahman et al. (2022b), weather modeling Bire et al. (2023); Pathak et al. (2022), robotics Bhaskara et al. (2023); Peng et al. (2023), and computer vision Guibas et al. (2021); Rahman & Yeh (2024); Viswanath et al. (2022). In our work, we propose a new parameterization for the graph neural operator to efficiently capture spatial dynamics in Lagrangian systems and generalize to different discretizations of reduced-order inputs.

**Reduced-order model**   Reduced-order models (ROMs) simplify high-dimensional dynamic systems by projecting them onto a lower-dimensional manifold, resulting in faster and less expensive computations (Berkooz et al., 1993; Holmes et al., 2012; Lee & Carlberg, 2020a; Peherstorfer, 2022). These methods gain computational efficiency by simulating a subset of the original spatial samples (An et al., 2008). Recent neural field-based ROMs (Pan et al., 2023; Yin et al., 2023; Wen et al., 2023; Chen et al., 2023) demonstrate the ability to train a discretization-agnostic low-dimensional representation, allowing the trained model to generalize over various geometric discretizations. We extend these ROMs by using a neural operator to compute the dynamics of the spatial samples, such that the model is a non-intrusive and discretization-agnostic ROM system. This combination creates a machine-learning model that generalizes across geometries while being significantly faster than traditional models.

## 3 METHOD: SPATIAL DIMENSION REDUCTION

In this section, we introduce the formulation for spatial-reduction and reduced-order representation of the input point-cloud. As shown in Figure 1., the input, represented as a point-cloud, is reduced into a sparse graph. We define the neural field formulation to recover the points not present within

the sparse graph. In this situation, discretization invariance refers to the model's agnosticism to the choice of points used in the sparse graph.

**Full-order system** Let $\{\boldsymbol{X}^j\}_{j=1}^P$ be the $P$-point discretization of the spatial domain $\Omega$, where $P$ is the number of full-order spatial points. Traditional full-order numerical PDE solvers directly operate on these spatial discretizations (Hughes, 2012) and are therefore prohibitively slow when $P$ is large.

**Reduced-order system** ROM techniques leverage a $Q$-point discretization of the spatial domain $\{\boldsymbol{X}^k\}_{k=1}^Q$, where $Q \ll P$. In particular, by leveraging neural fields and projection-based ROM, the work by Chang et al. (2023) proposes a technique that can infer the continuous spatial function at arbitrary spatial locations from just a few spatial samples, i.e., the $Q$-point discretization. To evolve these $Q$-point discretizations over time, we seek a mapping between $\{\phi_{t_n}^k\}_{k=1}^Q$ and $\{\phi_{t_{n+1}}^k\}_{k=1}^Q$, where $\phi_{t_n}^k = \mathbf{X}_{t_n}^k = \phi_{t_n}(\mathbf{X}^k)$.

**Sampling-based reduction** The Q-point discretization, $\{\boldsymbol{X}^k\}_{k=1}^Q \in \mathbb{R}^d$ is obtained by applying farthest point sampling on the $P$-point discretization of the system. This ensures an even distribution of points, reduced redundancy in closely clustered regions and preservation of geometric features. This system is then converted to a *sparse radius-graph*, connecting all the points in a neighborhood defined by a ball of radius $\mathbf{r}$. This process is illustrated in Figure 1.

**Time Integration** In the discrete setting, we leverage an explicit Euler time integrator (Ascher & Petzold, 1998) with step-size $\Delta t$,

$$\phi_{t_{n+1}}^j = \phi_{t_n}^j + \Delta t \, \dot{\phi}_{t_n}^j \tag{2}$$

$$\dot{\phi}_{t_{n+1}}^j = \dot{\phi}_{t_n}^j + \Delta t \, \ddot{\phi}_{t_n}^j \tag{3}$$

As such, the one and only unknown in the equation above is the acceleration $\mathbf{A}_{t_n}^j = \ddot{\phi}_{t_n}^j$, which is necessary for computing the velocity $\mathbf{V}_{t_{n+1}}^j = \dot{\phi}_{t_{n+1}}^j$. We propose to predict the acceleration field from the current and past velocity fields via neural operators to ensure discretization invariance (see Section 4).

**Full-order inference using neural fields** Equipped with the next time-step positions $\phi_{t_{n+1}}(\boldsymbol{X}^k)$ at the reduced-order $Q$-point discretizations $\{\boldsymbol{X}^k\}_{k=1}^Q$, we will compute the next time-step positions $\phi_{t_{n+1}}(\boldsymbol{X}^j)$ at the full-order $P$-point discretizations $\{\boldsymbol{X}^j\}_{j=1}^P$. To do so, we leverage a neural representation of projection-based reduced-order models (ROM) (Benner et al., 2015). ROM assumes that $\phi_{t_{n+1}}(\boldsymbol{X}^j)$ can be represented as a weighted sum of a small number of basis functions $\mathbf{U}$ with weights $\mathbf{q}_{t_{n+1}}$: $\phi_{t_{n+1}}(\boldsymbol{X}^j) = \boldsymbol{X}^j + \mathbf{U}(\boldsymbol{X}^j)\mathbf{q}_{t_{n+1}}$.

We emphasize that $\mathbf{U}$ is not restricted to a specific location in space and can be evaluated at any arbitrary point, making it independent of any particular discretization.

The basis functions are implemented using neural fields whose weights are learnable. We follow the same training procedure as described in Chang et al. (2023). After training, the basis $\mathbf{U}$ stays fixed over time while the weights $\mathbf{q}_{t_{n+1}}$ change at each time step.

When we have the function $\phi_{t_{n+1}}(\boldsymbol{X}^k)$ for the sub-sampling $\boldsymbol{X}^k \in \{\boldsymbol{X}^k\}_{k=1}^Q$, we can calculate $\mathbf{q}_{t_{n+1}}$ by solving the least squares problem:

$$\min_{\mathbf{q}_{t_{n+1}}} \sum_{k=1}^Q \|\phi_{t_{n+1}}(\boldsymbol{X}^k) - (\boldsymbol{X}^k + \mathbf{U}(\boldsymbol{X}^k)\mathbf{q}_{t_{n+1}})\|_2^2. \tag{4}$$

After $\mathbf{q}_{t_{n+1}}$ is obtained, we are able to calculate $\phi_{t_{n+1}}(\boldsymbol{X}^j)$ by:

$$\phi_{t_{n+1}}(\boldsymbol{X}^j) = \boldsymbol{X}^j + \mathbf{U}(\boldsymbol{X}^j)\mathbf{q}_{t_{n+1}}, \quad \forall \boldsymbol{X}^j \in \{\boldsymbol{X}^j\}_{j=1}^P. \tag{5}$$

# 4 METHOD: TEMPORAL DYNAMICS

The reduced-order representation of the system forms the backbone of the time-stepper model, which is represented in Figure 1. as the encoder-processor-decoder. In this section, we discuss how the

neural operator learns the time-stepping temporal dynamics of the reduced-order system represented by the sparse graph.

To learn the temporal dynamics, i.e., computing $\ddot{\phi}_{t_n}^j$ in Equation (3), we use a discretization-invariant neural operator architecture that follows the *encode-process-decode* setup. We propose a graph-based neural operator architecture called *Interaction Operator* as the encoder and the decoder, while we use a neural operator transformer (NOT) as the processor.

Neural operators are a class of machine learning models that learn to map functions in infinite-dimensional function spaces using a finite collection of discretized input-output pairs. In particular, we want to learn the mapping from the current and past velocity fields $(\dot{\phi}_{t_i})_{i=n-w}^n$ to the current acceleration field $\ddot{\phi}_{t_n}$. The hyperparameter $w$ defines the time window given by the past $w$ time steps. In practice, we use a finite collection of spatially discretized input-output pairs, as, e.g., provided by the $Q$-point discretization. However, importantly, the output of the neural operator will be consistent across different discretizations.

### 4.1 BACKGROUND: OPERATOR LEARNING

Many neural physics simulators model spatial interactions between particles using graph neural networks (GNNs). While applicable to different number of particles, GNNs struggle if there is a significant difference between training and inference sizes. To this end, we will use neural operators that are agnostic to the underlying resolution, i.e., the number of particles, *by construction*.

A graph neural operator (GNO) operates on a radius graph, where a point $x$ is connected to all points within a ball $B_r(x)$ of a certain radius $r$ Li et al. (2020b;c; 2024). This can be understood as a discretization of an integral transform

$$\text{GNO}(v)(x) = \int_{B_r(x)} \kappa_\theta(x, y, v(y))dy, \tag{6}$$

where $v$ denotes a suitable input function and $\kappa$ is a learnable kernel, parametrized by a neural network. If the input function $v$ is discretized at points $y_i \in B_r(x)$, the integral transform in equation 6 can be approximated by

$$\text{GNO}(v)(x) \approx \sum_{y_i \in B_r(x)} \kappa_\theta(x, y_i, v(y_i))\Delta y_i, \tag{7}$$

where $\Delta y_i$ are suitable integration weights.

While geometries in the physical domain are complex and irregular, we follow Li et al. (2024) and efficiently learn the global spatio-temporal dynamics on a coarse uniform grid in latent space. To switch between the given discretization and a uniform grid, we note that the output function $\text{GNO}(v)$ in equation 7 can be evaluated on points $x$ different from the discretization points $\{y_i\}_i$. We leverage this property to evaluate the output on a uniform grid, which is used as the latent space to learn the spatio-temporal dynamics.

### 4.2 ENCODING AND DECODING LOCAL SPATIAL FEATURES

To capture the local spatial features of the discretized input, we define a graph-based neural operator, termed *Interaction Operator*, which performs two tasks. It captures the point interactions using a discretization-agnostic adaptation of message passing and leverages a GNO layer to project the features to a regular grid. The general formulation of the message-passing operator is defined as

$$\text{MP}_k(v)(x) = f_\theta\Big(v(x), \int_{B_r(x)} \kappa_\theta(k(x, y), v(x), v(y))dy\Big). \tag{8}$$

In contrast to existing GNOs as in equation 6, we let the kernel $\kappa_\theta$ in equation 8 depend on $v(x)$ and an additional function $k$ representing edge features. Moreover, we allow for residual connections through $f_\theta$, which is parametrized by a neural network. The term $v(x)$ represents the local interactions between the nodes of the input graph. We can discretize equation 8 similar to equation 7 but require the evaluation point $x$ to be included in the set of discretization points $\{y_i\}_i$ at which we know the value of the input function $v$ and the edge features $k$. To this end, we use the same discretization for the input and output functions of the message-passing operator. To be able to use a uniform

discretization in the latent space, we define the interaction operators as compositions with GNO layers as in equation 6, i.e., $\text{IO}^{\text{enc}} = \text{GNO} \circ \text{MP}_h$ and $\text{IO}^{\text{dec}} = \text{MP}_k \circ \text{GNO}$.

For the edge features $h$ and $k$, we choose $h(x, y) = g_\theta((x - y)/r)$ and $k(x, y) = h(x, y) + \kappa_\theta(h(x, y), v(x), v(y))$, where $g_\theta$ is parametrized by a neural network and $\kappa_\theta$ is the kernel of $\text{MP}_h$. This effectively creates a residual connection between the interaction operators. In summary, using the interaction operators $\text{IO}^{\text{enc}}$ and $\text{IO}^{\text{dec}}$ as encoder and decoder allows us to map from an arbitrary discretization of the input and output fields to a uniform discretization in the latent space.

### 4.3 GLOBAL SPATIO-TEMPORAL PROCESSING

To learn the spatio-temporal evolution of the system in the latent space, we use a neural operator transformer (NOT). The transformer can be viewed as a sequence of global GNO layers as in equation 6 with a specific choice of kernel Kovachki et al. (2023). It processes the output of the interaction operator, which is a function discretized on a coarse regular grid. These inputs are first transformed to embeddings through pointwise MLPs. Then, heterogeneous attention blocks as proposed in Hao et al. (2023) are used to compute the normalized self-attention between the embeddings. The overall architecture of the neural operator $\mathcal{G}_\theta$ mapping the past velocity fields to the current acceleration field can then be defined as $\mathcal{G}_\theta(v) = \text{IO}^{\text{dec}} \circ \text{NOT} \circ \text{IO}^{\text{enc}}$

The pseudocode is provided in Appendix D. To summarize, the model learns the instantaneous acceleration, denoted as $\mathbf{A}_{t_n} = \mathcal{G}_\theta(\mathbf{V}_{t_{n-w:n}})$, where, $w$ is the window used for past time step instances and $\mathbf{V}_{t_{n-w:n}}$ denotes the velocity sequence.

## 5 EXPERIMENTS

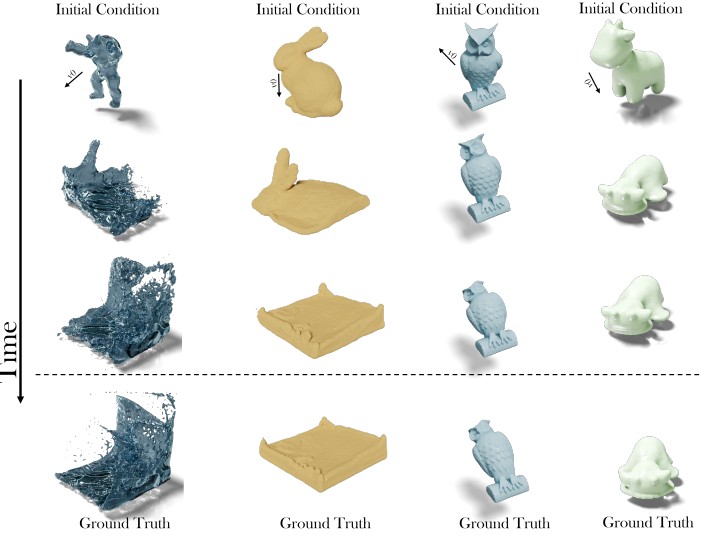

Figure 2: **Performance against different systems**: The figure shows the full-order rollout performance on Water, Sand, Elasticity and Plasticine

### 5.1 DATASET

We trained our model on four 3-dimensional physical systems - Newtonian fluids (Water), Drucker-Prager elastoplasticity (Sand), von Mises yield (Plasticine) and purely Elastic deformations. We assume that all these materials follow the elastodynamic equation, given by

$$\rho_0 \ddot{\phi} = \nabla \cdot \mathbf{P} + \rho_0 \mathbf{b} \tag{9}$$

where, $\mathbf{P}$ is the first Piola-Kirchoff stress, $\rho_0$ is the initial density, $\mathbf{b}$ is the body force and $\phi$ is the deformation map.

We used the nclaw simulator (Ma et al., 2023) to generate 100 trajectories for each of these systems with random initial velocity conditions and a fixed boundary $[0, 1], [0, 1], [0, 1]$, with a free-slip boundary condition. The $\Delta t$ between consecutive time frames was $5e^{-3}$s. We additionally trained our model on four 2-D systems provided by Sanchez-Gonzalez et al. (2020) - WaterDrop, Sand, Goop and MultiMaterial.

## 5.2 Model setup and Hyperparameters

**Data Representation**  To train the time-stepper model, we create a window of $w$ point cloud position sequences as the input, with the pointwise acceleration as the output. We define $\mathbf{X}_{t_n} \in \mathbb{R}^{Q \times d}$ to be the pointwise positions of $Q$ particles within a $d$-dimensional system at time $n$. A sequence of $N$ time steps is denoted as $\mathbf{X}_{t_{0:N}} = (\mathbf{X}_{t_0}, \dots, \mathbf{X}_{t_N})$. In particular, $\{\mathbf{X}_{t_n}^0, \dots, \mathbf{X}_{t_n}^Q\} \in \mathbf{X}_{t_n}$ are the individual particles within the system. We define velocity at time $n$ as $\mathbf{V}_{t_n} \in \mathbb{R}^{Q \times d}$ as $\mathbf{X}_{t_n} - \mathbf{X}_{t_{n-1}}$. Similarly, acceleration at time $n$ is defined as $\mathbf{A}_{t_n} = \mathbf{V}_{t_n} - \mathbf{V}_{t_{n-1}}$ or $\mathbf{A}_{t_n} = \mathbf{X}_{t_{n+1}} - 2\mathbf{X}_{t_n} + \mathbf{X}_{t_{n-1}}$. In all these cases, $\Delta t$ is set to one for simplicity. In case of water and sand, the velocity sequence is perturbed with noise. The particle types (water, sand, plasticine, etc.) are represented as embeddings.

**Boundary Representation**  To enforce the boundaries of the system, the node feature includes the past $w$ velocity fields as well as the distance of the most recent position field to the upper ($b_u$) and lower ($b_l$) boundaries of the computational domain, given by $\mathcal{D} = [(x_i - b_l)/r, (b_u - x_i)/r]$, where $r$ is the radius of the graph.

**Sampling and Graph Construction**  To reduce the point cloud to ROM space, we use farthest point sampling to achieve an even spatial distribution of points. These sampled points are represented as the vertices of a radius graph, whose neighbors are defined as the points within the specified radius. The radius is tuned to ensure that the reduced-order graph has the same number of components as the full-order graph. We show that if the number of components increases, it leads to unphysical volume collapse. These effects are shown in Figure 3 and Table 13 in Appendix G.1.

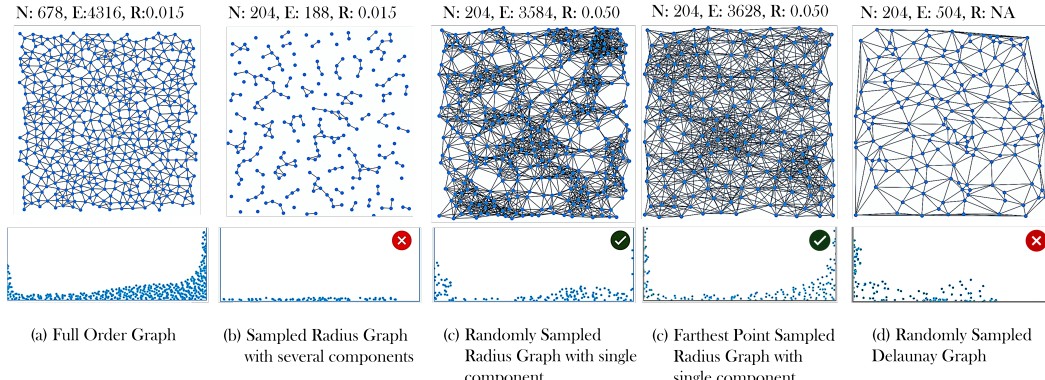

| N: 678, E:4316, R:0.015 | N: 204, E: 188, R: 0.015 | N: 204, E: 3584, R: 0.050 | N: 204, E: 3628, R: 0.050 | N: 204, E: 504, R: NA |

(a) Full Order Graph — (b) Sampled Radius Graph with several components — (c) Randomly Sampled Radius Graph with single component — (c) Farthest Point Sampled Radius Graph with single component — (d) Randomly Sampled Delaunay Graph

Figure 3: **Our method operates on a reduced-order graph.** **(a)** depicts the graph with all the points. **(b)** When the reduced-order graph has more components than the full-order graph, it leads to volume collapse in simulations. **(c)** We improve the prediction of the dynamics by choosing a radius that induces the same number of components. **(d)** FPS based sampling has similar performance but the system is more uniformly distributed. **(e)** Delaunay Graph causes the system to break. (N: No. of Nodes, E: No. of Edges, R: Graph Radius)

**Loss Function**  The time stepper model predicts per-particle acceleration from a sequence of past velocities of the samples. The loss is defined as the mean squared error between the predicted acceleration and the ground-truth acceleration in the simulation sequence. To account for the impact of noise and normalization, we compute the weighted average between acceleration loss and the

MSE on the predicted and the expected positions. We choose a large $\beta$ in the order of $1e^5$. For a consecutive pair of positions $\mathbf{X}_{t_n}$ and $\mathbf{X}_{t_{n+1}}$, with corresponding velocities $\mathbf{V}_{t_n}$ and $\mathbf{V}_{t_{n+1}}$, the corresponding acceleration is defined as $\hat{\mathbf{A}}_{t_n} = (\mathbf{V}_{t_{n+1}} - \mathbf{V}_{t_n})/\Delta t$. The loss is thus given as

$$L(\theta) = \|\mathcal{G}_\theta(\mathbf{V}_{t_{n-w:n}}) - \hat{\mathbf{A}}_{t_n}\|_2^2 + \beta\|\hat{\mathbf{X}}_{t_{n+1}} - \mathbf{X}_{t_{n+1}}\|_2^2 \tag{10}$$

The neural field is trained using the reconstruction loss (Chang et al., 2023), given by

$$L(\theta) = \sum_{n=0}^{N}\sum_{j=1}^{P} \|\boldsymbol{X}^j + \mathbf{U}_\theta(\boldsymbol{X}^j)\mathbf{q}_{t_{n+1}} - \phi_{t_{n+1}}^{GT}(\boldsymbol{X}^j)\|_2^2 \tag{11}$$

and $\phi_{t_{n+1}}^{GT}$ is the ground truth deformation map at time $t_{n+1}$ and at spatial sample $\boldsymbol{X}^j$. Additional training details and model hyperparameters can be found in Appendix C.1.

## 5.3 RESULTS

**Performance on different physical systems** We evaluate our model on previously unseen trajectories of different physical systems (validation dataset) in Table 1, which presents the mean-squared error MSE loss over one time-step and over several time-steps accumulated auto-regressively ("rollout"). The loss is computed on the position vector, which is computed by applying Euler integration on the model generated acceleration vector. Figure 2 visually depicts the outputs rolled out by the model. Figure 5b shows the model-generated point-clouds, where the points shown are the spatial locations of the system. Figure 4 depicts the performance w.r.t the ground truth point clouds in 2D settings.

Table 1: This table showcases the performance of GIOROM on several physical systems, These results are computed on the full-order system.

| PHYSICAL SYSTEM | DURATION ($5e^{-3}$s) | # POINTS | SPARSE GRAPH SIZE | SCALE | NOISE | ONE STEP-MSE ($\times e^{-9}$) | ROLLOUT MSE ($\times e^{-3}$) |
|---|---|---|---|---|---|---|---|
| WATER-3D | 1000 | 55k | 1.7k | 32× | $3e^{-4}$ | 5.23 | 0.386 |
| WATER-2D | 1000 | 1k | 0.12k | 8.3× | 0 | 0.524 | 6.7 |
| SAND-3D | 400 | 32k | 1k | 32× | $3e^{-7}$ | 4.87 | 0.0025 |
| SAND-2D | 320 | 2k | 0.3k | 6.6× | 0 | 8.5 | 1.34 |
| GOOP-2D | 400 | 1.9k | 0.2k | 9.5× | 0 | 1.31 | 0.94 |
| PLASTICINE | 320 | 5k | 1.1k | 4.5× | 0 | 0.974 | 0.5 |
| ELASTICITY | 120 | 78k | 2.6k | 30× | 0 | 0.507 | 0.2 |
| MULTI-MATERIAL 2D | 1000 | 2k | 0.25k | 8× | 0 | 2.3 | 9.43 |

Table 2: This table highlights resolution invariance and discretization invariance of GIOROM in different settings of the Elasticity dataset.

| SETTING | AVERAGE NUM. POINTS | SCALE W.R.T TRAINING DATA | ONE-STEP MSE (x $e^{-9}$) | ROLLOUT MSE (x $e^{-3}$) |
|---|---|---|---|---|
| DIFFERENT DISC. | 2.5k | 1.25 | 0.8 | 0.2 |
| LOWER RES. | 1k | 0.5 | 1.9 | 0.5 |
| LOWER RES. | 0.5k | 0.25 | 2.34 | 0.6 |
| HIGHER RES. | 5k | 2 | 0.319 | 0.7 |
| HIGHER RES. | 10k | 4 | 0.88 | 0.9 |
| DIFFERENT GEOMETRY | 98k | 32 | 10.7 | 5.7 |
| FULL ORDER INFERENCE WITH IO | 78k | 52 | 94.4 | 2 |

**Discretization invariance** We evaluate the discretization invariance through the experiments presented in Table 2. These were performed on the elasticity dataset, due to its full-order size of 78k particles. The first row shows the performance on a validation dataset, measured as the MSE between Euler integrated positions and the expected positions. Each input comprises 1.2× the number of points used in the corresponding training dataset. This ensures that the input has comparable resolution but different spatial instantiations of the same input. However, we test on previously unseen trajectory (inital condition) in all of these cases We perform two sets of experiments on lower

resolution inputs ($0.25\times$ and $0.5\times$) and higher ones ($2\times$, $4\times$). We also test generalization to unseen geometries, as shown in the sixth row of Table 2, and lastly, to justify the need for neural field, we infer the full-order system using the time-stepper and observe a slight degradation in performance compared to the neural field. Discretization invariance is illustrated in Figure 5a.

## 5.4 BASELINES

**ROM baselines**    We evaluate our proposed neural field against Proper Orthogonal Decomposition (POD) and MLP based autoencoder models similar to those proposed in Lee & Carlberg (2020b). We observe that when the discretization is changed, these models struggle to infer the spatial locations of the system. However, our approach is agnostic to the spatial indices of the sampled system. On randomized sub-samples of the same input point-cloud from the elasticity dataset (78k points), we observed that POD had an MSE of 6.00e-4, while the Autoencoder had an MSE of 2.10e-4. Our model achieved an MSE of 7.59e-7, highlighting discretization invariance.

**Neural Operator baselines**    Table 3 represents the rollout performance of different Neural Operator models on reduced-order graphs. The performance is measured as the average MSE accumulated over the entire duration. We compare against **GINO** Li et al. (2024), General Neural Operator Transformer **GNOT** Hao et al. (2023) and Inducing Point Operator Transformer Lee & Oh (2024). Additionally, we compare against two graph neural network based models **GAT**, **GNN**, similar to the model proposed in Sanchez-Gonzalez et al. (2020).

Table 3: This table compares the rollout MSE of GIOROM time-stepper against other neural physics solvers. These results were computed on the reduced-order system, which is the training setting for all these models

| MODEL | WATER-3D | PLASTICINE | ELASTIC | SAND-3D |
|-------|----------|------------|---------|---------|
| GNN   | 0.011    | 0.0038     | 0.0019  | 0.0008  |
| GAT   | 0.06     | 0.0083     | 0.0097  | 0.011   |
| GINO  | 0.38     | 0.09       | 0.18    | 0.07    |
| GNOT  | 0.046    | 0.0052     | 0.0028  | 0.0085  |
| IPOT  | 0.15     | 0.097      | 0.084   | 0.0075  |
| **OURS** | 0.0106 | 0.0008    | 0.0004  | 0.0009  |

## 6 DISCUSSION

**Architecture Choice**    The architecture contains 3 elements - the Interaction Operator, the Neural Operator Transformer and the Neural Field. To underscore the importance of each of these components, we perform several ablations. In the absence of the Interaction Operator, the model fails to capture local spatial interactions effectively. The Neural Operator Transformer ensures that the model can generalize to longer trajectories, without requiring the velocity to be injected with noise, unless the system is highly dynamic, as in the case of 3D water and sand simulations. We present the ablations in Table 4. As shown in tables 2 and 9, the use of neural fields speeds up the inference, however, it doesn't significantly improve the accuracy.

**Compatibility With Various Neural Physics solvers**    The core aspect of our framework is that it enables learning physics in a reduced order setting, allowing for inference at any spatial point with arbitrary resolution or discretization. Besides the Interaction Operator and Neural Operator Transformer integrated into our model, other discretization-agnostic methods for learning temporal dynamics are also applicable. As illustrated in Table 5, our setup achieves strong performance when substituting the Neural Operator Transformer with a Fourier Neural Operator, though the inclusion of noise during training is necessary for all physical systems. Additionally, GNS Sanchez-Gonzalez et al. (2020) can be utilized as a time stepper, but the computational speed decreases due to the ten message-passing blocks.

**Justification of Neural Fields as a key factor in achieving speedup**    In Table 10., we highlight how the inference time increases with the increase in the size of the input graphs. To overcome this

Table 4: This table experimentally shows the importance of each component within the architecture. These numbers were computed on the Plasticine dataset. It can be observed, that NOT reduces the dependency on noise, while IO improves the accuracy

| SETUP | NOISE | 1-step MSE | Rollout MSE |
|---|---|---|---|
| **Ours** | **0** | **1.17e-9** | **0.0008** |
| NOT w/o IO | 0 | 3.35e-9 | 0.05 |
| IO + FNO + IO | 0 | 2.1e-9 | 0.117 |
| IO + FNO + IO | 3e-4 | 3.4e-9 | 0.0032 |
| 2 GNO + FNO + 2 GNO | 3e-4 | 1.9e-7 | 0.09 |
| GNO + FNO + GNO | 0 | 8.0e-9 | 21.84 |
| GNO + FNO + GNO | 3e-6 | 9.5e-9 | 16.70 |
| GNO + FNO + GNO | 3e-4 | 2.8e-7 | 0.36 |
| GNO + FNO + GNO | 3e-3 | 2.67e-5 | 2.85 |

Table 5: This table shows the performance of different neural physics solvers as the time stepper. The time complexity of GNS and FNO's dependency on training noise are the two tradeoffs that were considered while choosing our architecture.

| TIME STEPPER | WATER-3D | SAND-3D | PLASTICINE | ELASTICITY |
|---|---|---|---|---|
| OURS | 0.0106 | 0.0009 | 0.0008 | 0.0004 |
| GNS | 0.011 | 0.0008 | 0.0038 | 0.0019 |
| IO + FNO + IO | 0.025 | 0.0067 | 0.0072 | 0.0058 |

bottleneck, we propose using smaller graphs for time-stepping and the neural field to recover the full-order system. The neural field exhibits a near constant time complexity across different sizes of the input point cloud. This is empirically shown in Table 5, where the upscale time of the neural field is nearly the same for different densities of full-order systems.

**Computation of Neural Field weights in Practice**    Equation 4 presents the least-squares expression for computing the weights $q_{t_{n+1}}$. In practice, this is formulated as solving a symmetric positive linear system using a single Cholesky factorization, as shown in Chang et al. (2023). Therefore, this does not include expensive computation overheads. This is shown in Table 9.

**Handling self-contact in Materials**    The training data for the model is generated using MPM solvers, which do not explicitly check for self-collision, but rather implicitly handle them through a background grid, for both solids and fluids. Being data-driven, this phenomenon is learned by the model implicitly. Better fine-grained self-contact sampling is an exciting future work direction.

# 7    CONCLUSION

In conclusion, our proposed GIOROM, can implicitly learn PDEs over several physical systems. Utilizing a reduced-order modeling approach on sparse graphs, GIOROM is faster than previous neural network-based physics solvers while achieving high fidelity simulations. Moreover, our neural-operator-based model generalizes well across different initial conditions, velocities, discretizations, and geometries.

Despite its promising performance, GIOROM has limitations that warrant further exploration. While GIOROM is capable of generalizing across different settings, like many machine learning and reduced-order methods, it struggles with extreme out-of-distribution scenarios (Li et al., 2020b; Chen et al., 2023). Moreover, while GIOROM is primarily designed for continuous systems, future research might explore mechanisms to explicitly handle discontinuities (Belhe et al., 2023; Goswami et al., 2022).

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

## A  ADDITIONAL RELATED WORKS

**Time series dynamical systems**  Simulating temporal dynamics in an auto-regressive manner is a particularly challenging task due to error accumulations during long rollout Wikner et al. (2024); List et al. (2024). There have been many works that learn temporal PDEs and CFD, including Majid & Tudisco (2024); Liu et al. (2024c); Sarkar et al. (2024); Wu et al. (2024); Jeon et al. (2024); Jiang et al. (2024); Ma et al. (2024); Janny et al. (2024). Some works have proposed neural network-based approaches to model 3D Lagrangian dynamics, such as Ummenhofer et al. (2020), who propose a convolutional neural network-based approach to model the behavior of Newtonian fluids in 3D systems. Sanchez-Gonzalez et al. (2020) propose a more general graph-based framework, but the network suffers from high computation time on very dense graphs and is restricted to learning physics in the full-order setting.

## B  OPERATOR LEARNING

### B.1  BACKGROUND

Here, we summarize the important ingredients of neural operators. For more details, please refer to Li et al. (2020a). Operator learning is a machine learning paradigm where a neural network is trained to map between infinite-dimensional function spaces. Let $\mathcal{G} : \mathcal{V} \to \mathcal{A}$ be a nonlinear map between the two function spaces $\mathcal{V}$ and $\mathcal{A}$. A neural operator is an operator parameterized by a neural network given by

$$\mathcal{G}_\theta : \mathcal{V} \to \mathcal{A}, \quad \theta \in \mathbb{R}^P, \tag{12}$$

that approximates this function mapping in the finite-dimensional space. The learning problem can be formulated as

$$\min_{\theta \in \mathbb{R}^P} \mathbb{E}_{v \sim D} \left[ \|\mathcal{G}_\theta(v) - \mathcal{G}(v)\|_{\mathcal{V}}^2 \right], \tag{13}$$

where $\|\cdot\|_{\mathcal{V}}$ is a norm on $\mathcal{V}$ and $D$ is a probability distribution on $\mathcal{V}$. In practice, the above optimization is posed as an empirical risk-minimization problem, defined as

$$\min_{\theta \in \mathbb{R}^P} \frac{1}{N} \sum_{i=1}^{N} \|\mathcal{G}_\theta(v^{(i)}) - a^{(i)}\|_{\mathcal{V}}^2. \tag{14}$$

A neural operator $\mathcal{G}_\theta$ learns the mapping between two functions through a sequence of point-wise and integral operators, defined as

$$\mathcal{G}_\theta = \mathcal{L} \circ \mathcal{J}_1 \circ ... \circ \mathcal{J}_L \circ \mathcal{P} \tag{15}$$

The lifting and projection layers $\mathcal{L}$ and $\mathcal{P}$ are learnable pointwise operators that output a function with a higher and lower-dimensional co-domain, respectively. The intermediate layers $\mathcal{J}_\ell$ perform kernel integration operations with a learnable kernel function as in equation 6.

## C  INTERACTION NETWORK

The interaction network proposed in Battaglia et al. (2016) learns a relation-centric function $f$ that encodes spatial interactions between the interacting nodes within a system as a function of their interaction attributes $r$. This can be represented as

$$e_{t+1} = f_R(x_{1,t}, x_{2,t}, r) \tag{16}$$

A node-centered function predicts the temporal dynamics of the node as a function of the spatial interactions as follows

$$x_{1,t+1} = f_o(e_{t+1}, x_{1,t}) \tag{17}$$

In a system of $m$ nodes, the spatial interactions are represented as a graph, where the neighborhood is defined by a ball of radius r. This graph is represented as $G(O, R)$, where $O$ is the collection of objects and $R$ is the relationships between them. The interaction between them is defined as

$$\mathcal{I}(G) = f_o(a(G, X, f_R(\langle x_i, x_j, r_{ij} \rangle))) \tag{18}$$

Where $a$ is an aggregation function that combines all the interactions, $X$ is the set of external effects, not part of the system, such as gravitational acceleration, etc.

## C.1 Hyperparameters

The models were implemented using `Pytorch` library and trained on `CUDA`. The graphs were built using `Pytorch Geometric` module. All models were trained on `NVIDIA RTX 3060` GPUs for 5e6 steps.

The input to the model is a state vector matrix corresponding to $w = 6$ previous time steps of each particle, along with features that represent the material of each particle. A radius graph is constructed for the set of particles within the input space, such that edges are added between particles that are within the radius $r$. The nodes of the graph are the velocity sequences for all the particles within the sparse graph.

The graph is constructed using `radius_graph` defined in `Pytorch Geometric`. The node features and the edge features, which include the distance from the boundary points, are encoded into latent vectors of size 128 using 2 MLPs. The encoder uses two layers of interaction operator. The latents are then processed by two layers of Neural Operator Transformer. The decoder layers are symmetric to the encoder layers. However, the decoder uses an additional projection layer with 16 channels that lifts the output to 128 channels, which is then projected back to the physical dimensions of the input graph (2D or 3D). All MLPs within the `GNO` and `FNO` framework use `gelu` activation function.

**Training noise**    In more dynamic systems such as `Water-3D` and `Sand-3D`, to prevent noise accumulation during long rollouts, the velocity sequence is corrupted with random walk noise during training. The noise is sampled from a normal distribution $\mathcal{N}(0, \sigma^2)$. Systems like Plasticine or Elasticity did not require any training noise.

**Normalization**    All velocity sequences are standardized to zero mean and unit standard deviation. The dataset statistics are computed during training. Global mean and variance values from the training dataset are used to compute statistics.

**Optimizers**    Optimization is done with Adamax optimizer, with an initial learning rate of 1e-4, weight decay of 1e-6 and a batch size of 4. The learning rate was decayed exponentially from $10^{-4}$ to $10^{-6}$ using a scheduler, with a gamma of $0.1^{1/5e6}$

## D    Pseudocode

---
**Algorithm 1** Predicting Lagrangian dynamics with GIOROM

---
**Input:** Reduced-order velocities $\mathbf{V}_{t_{n-w:n}} = \{\mathbf{V}_{t_{n-w:n}}^k\}_{k=1}^Q$, full-order points $\bar{\mathbf{X}} = \{\mathbf{X}^j\}_{j=1}^P$
**Output:**  Full-order deformation $\bar{\mathbf{X}}_{t_{n+1}} = \{\mathbf{X}_{t_{n+1}}^j\}_{j=1}^P = \{\phi_{t_{n+1}}(\mathbf{X}^j)\}_{j=1}^P$

1: $\mathbf{A}_{t_n} \leftarrow \mathsf{G}_\theta\left(\mathbf{V}_{t_{n-w:n}}\right)$                               ▷ See Section 4
2: $\mathbf{V}_{t_{n+1}} \leftarrow \mathbf{V}_{t_n} + \Delta t\, \mathbf{A}_{t_n}$                       ▷ See Equation equation 3
3: $\mathbf{X}_{t_{n+1}} \leftarrow \mathbf{X}_{t_n} + \Delta t\, \mathbf{V}_{t_{n+1}}$                   ▷ See Equation equation 2
4: $\bar{\mathbf{X}}_{t_{n+1}} \leftarrow \mathtt{NeuralField}(\mathbf{X}_{t_{n+1}}, \bar{\mathbf{X}})$       ▷ See Section 3

---

## E    Additional dataset details

We model the following classes of materials - elastic, plasticine, granular, Newtonian fluids, non-Newtonian fluids, and multi-material simulations.

**Plasticine (von Mises Yield)**    Using the `NCLAW` simulator, we generated 100 trajectories of 400 time steps (dt = $5e-4$) with random initial velocities and 4 different geometries - Stanford bunny, Stanford armadillo, blub (goldfish), and spot (cow). The trajectories are modeled using Saint Venant-Kirchoff elastic model, given by

$$\mathbf{P} = \mathbb{U}(2\mu\epsilon + \lambda tr(\epsilon))\mathbb{U}^T \tag{19}$$

---

**Algorithm 2** Training of the neural operator

---

**Input:** Reduced-order position sequence $\mathbf{X}_{t_{n-w:n}}$, ground truth acceleration $\hat{\mathbf{A}}_{t_n}$
**Output:** Reduced-order acceleration $\mathbf{A}_{t_n}$

1: $\mathbf{V}_{t_{n-w:n}} \leftarrow (\mathbf{X}_{t_{n-w:n}} - \mathbf{X}_{t_{n-w:n-1}})/\Delta t$
2: $\tilde{\mathbf{V}}_{t_{n-w-1:n}} \leftarrow \mathbf{V}_{t_{n-w:n}} + \mathcal{N}(0, \sigma^2)$ $\qquad\qquad$ ▷ As explained in Section C.1
3: edges $\leftarrow$ `radius_graph`($\mathbf{X}_{t_n}$, radius)
4: edge_feats $\leftarrow$ `MLP`($\mathbf{X}_{t_n}$, edge)
5: node_feats $\leftarrow$ `MLP`($\tilde{\mathbf{V}}_{t_{n-w:n}}$)
6: node_feats, edge_feats $\leftarrow$ `MP`(node_feats, edge_feats) ▷ `Message Passing` As in equation 8
7: lgrid $\leftarrow$ `linspace` ([min, max])
8: latents $\leftarrow$ `IT`($\mathbf{X}_{t_n}$, lgrid, node_feats) $\qquad\qquad$ ▷ `Integral Transform` See equation 7
9: acc $\leftarrow$ `NOT` (latents)
10: acc_spatial $\leftarrow$ `IT` (lgrid, $\mathbf{X}_{t_n}$, acc)
11: $\mathbf{A}_{t_n} \leftarrow$ `MP` (acc_spatial, edge_feats)
12: loss $\leftarrow$ `MSE`($\mathbf{A}_{t_n}$, $\hat{\mathbf{A}}_{t_n}$)

---

where $\lambda$ and $\mu$ are Lamé constants, $\mathbf{P}$ is the second Piola-Kirchoff stress and $\epsilon$ is the strain. $\mathbb{U}$ is obtained by applying SVD to the deformation gradient $\mathbf{F} = \mathbb{U}\mathbf{\Sigma}\mathbf{V}^{\mathbf{T}}$. The von Mises yield condition is denoted by

$$\delta\gamma = \|\hat{\epsilon}\| - \frac{\tau_Y}{2\mu} \tag{20}$$

where $\epsilon$ is the normalized Henky strain, $\tau_Y$ is the yield stress.

**Granular material (Drucker Prager sand flows)** We trained the model on 2 datasets to simulate granular media. We generated 100 trajectories at 300 time steps, using `NCLAW` simulator and on the 2D Sand dataset released by Pfaff et al. (2020). The Drucker-Prager elastoplasticity is modeled by the same Saint Venant–Kirchhoff elastic model, given by Equation 19. Additionally, the Drucker-Prager yield condition is applied such that

$$tr(\epsilon) > 0 \quad or \quad \delta\gamma = \|\hat{\epsilon}\| + \alpha\frac{(3\lambda + 2\mu)tr(\epsilon)}{2\mu} > 0 \tag{21}$$

where, $\alpha = \sqrt{2/3}\frac{2sin\theta}{3-sin\theta}$ and $\theta$ is the frictional angle of the granular media.

**Elasticity** To simulate elasticity, we generated simulations using meshes from Thingi10k dataset Zhou & Jacobson (2016). We generated 24 trajectories, with 200 time steps, for 6 geometries to train the model. The elasticity is modeled using stable neo-Hookean model, as proposed in Smith et al. (2018). The energy is denoted by

$$\Psi = \frac{\mu}{2}(I_C - 3) + \frac{\lambda}{2}(J - \alpha)^2 - \frac{\mu}{2}log(I_C + 1) \tag{22}$$

where $I_C$ refers to the first right Cauchy-Green invariant and $J$ is the relative volume change. $\mu$ and $\lambda$ are Lamé constants. The corresponding Piola-Kirchoff stress is given by

$$\mathbf{P} = \mu\Big(1 - \frac{1}{I_C + 1}\Big)\mathbf{F} + \lambda(J - \alpha)\frac{\partial J}{\partial \mathbf{F}} \tag{23}$$

where $\mathbf{F}$ is the deformation gradient.

**Newtonian Fluids** For Newtonian fluids, In the 2D setting, we use `WaterDrop` dataset created by Pfaff et al. (2020), which is generated using the material point method (MPM). For the 3D setting, we generated 100 trajectories with random initial velocity, each spanning 1000 time steps at a dt of $5e - 3$. This dataset was prepared using the `NCLAW` framework. These are modeled as weakly compressible fluids, using fixed corotated elastic model with $\mu = 0$. The Piola-Kirchoff Stress is given by

$$\mathbf{P} = \lambda J(J - 1)\mathbf{F}^{-T} \tag{24}$$

**Non-Newtonian fluids**  To train the model on non-Newtonian fluids, we used the `Goop` and `Goop-3D` datasets.

**Multimaterial**  We simulated multi-material trajectories in 2D using the dataset published by Pfaff et al. (2020).

## F  TRAINING DETAILS

Ground truth acceleration is computed from position sequences before adding noise to the input (in case of systems that required training noise). This is adjusted by removing the velocity noise accumulated at the end of the random walk. This ensures that the model corrects the noise present in the velocity.

It is to be noted that this loss is defined as a 1-step loss function over a pair of consecutive time steps $k$ and $k + 1$, imposing a strong inductive bias towards a Markovian system. Optimizing the model for rollout over $K$ steps would overlook the effects of instantaneous physical states (influence of gravity, etc.), thus resulting in greater one-step errors, which would eventually accumulate and result in larger errors during rollout.

The model was validated by full rollouts on 10 held-out validation sets per material simulation, with performance measured by the MSE between predicted particle positions and ground-truth particle positions.

We test our model on multiple materials, ranging from Newtonian fluids to elastic solids, in both 2D and 3D settings. We empirically show that our model is at least 2-4x faster than graph neural network-based solvers with comparable parameter counts on the same simulation trajectory. Furthermore, we show that this speed-up doesn't compromise the accuracy of rollout predictions. We also highlight the generalization capability of our model to unseen initial trajectories and graph densities.

Table 6: The table denotes the various training and testing geometries.

| SHAPE PARAMS | ARMADILLO | BUNNY | SPOT | BLUB |
|---|---|---|---|---|
| MEAN CURVATURE | 2.9e-3 | 1.1e-2 | 1.3e-2 | 5.9e-3 |
| DIRICHLET ENERGY | 2.3e-4 | 2.5e-3 | 2.4e-3 | 8.1e-4 |

**Evaluation**  The evaluation metrics used to evaluate the models are particle-wise one-step MSE and rollout MSE on the held-out evaluation sets. The rollout velocity and positions are computed using semi-implicit Euler integration as

$$\mathbf{V}_{t_{k+1}} = \frac{\Delta \mathbf{X}_{t_k}}{\Delta t} + \Delta t \cdot \mathcal{G}_\theta(\mathbf{V}_{t_{k-C:k}}) \tag{25}$$

$$\mathbf{X}_{t_{k+1}} = \mathbf{X}_{t_k} + \Delta t \cdot \mathbf{V}_{t_{k+1}} \tag{26}$$

In our calculations, we assume $\Delta t$ to be 1.

## G  ABLATIONS

**Speedup against graph neural networks**  Graph neural networks can effectively capture spatial interactions in point clouds. However, the message passing operation adds a computational overhead that we overcome with neural operator layers. We show, in Table 7 and Table 8, that our model has faster inference times compared to graph based neural networks.

**Generalizability to degree of sparsity**  We tested the model against different degrees of sparsity, while maintaining the number of connected components, with respect to the full-order system. We observed, that the model performed consistently when the system was super-sampled, but the performance degraded when the system had fewer than 375 points or $0.25\times$ the average training data size. The results are visualized in Figure 6.

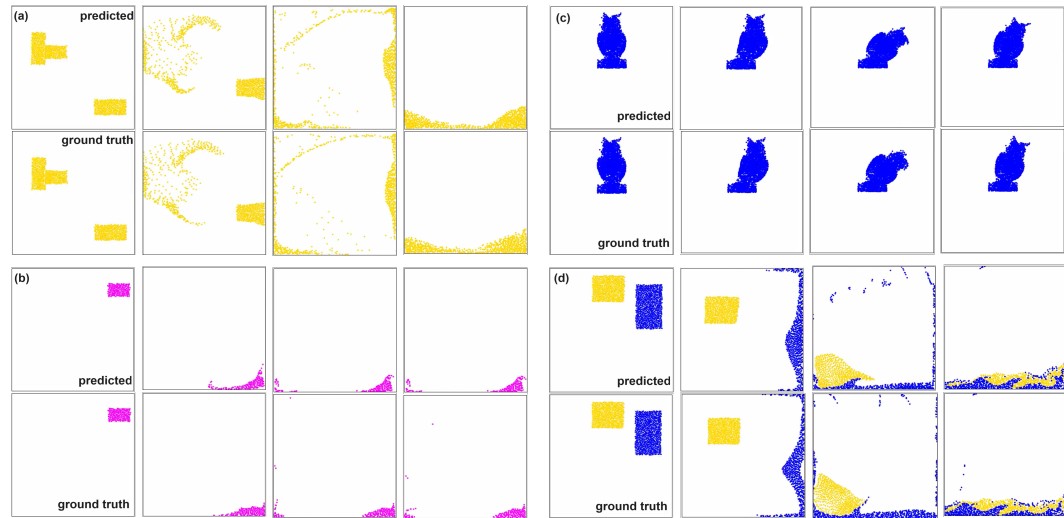

Figure 4: **The above figures depict full-order inference by GIOROM on 2D point clouds**. **(a)** depicts granular flow, **(b)** represents the trajectory of jelly-like substance under gravity. **(c)** shows the effects of external force on a highly elastic object. **(d)** depicts the interaction of granular media and Newtonian fluid.

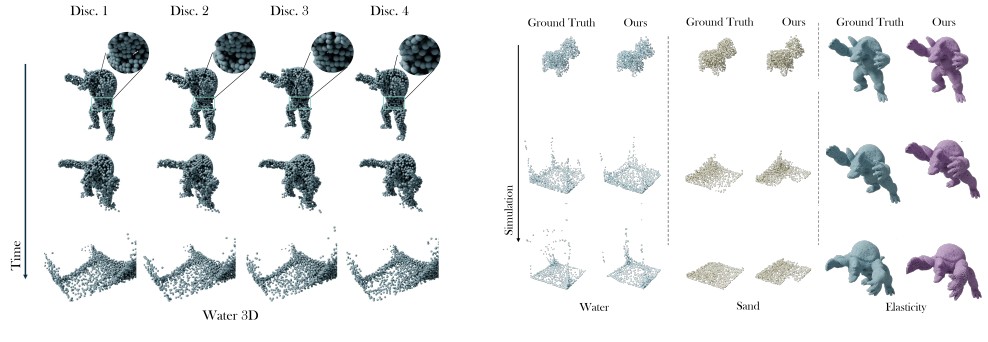

(a) **Discretization agnosticism.**          (b) **Point-cloud outputs from the time-stepper**

Figure 5: **Discretization Invariance and visualization of different physical systems inferred by the model.**

Table 7: Contrasting the change in computation time with the increase in connectivity radius for a graph with 7056 points. The times shown represent the overall time needed to infer all 200 time steps. We compare our time-stepper with other neural network based physics solvers.

| MODEL | TIME STEPS | NUMBER OF SPATIAL POINTS | CONNECTIVITY RADIUS | | | | | | |
|---|---|---|---|---|---|---|---|---|---|
| | | | 0.040 | 0.050 | 0.060 | 0.070 | 0.080 | 0.090 | 0.100 |
| OURS | 200 | 7056 points | **20.1s** | **34.3s** | **47.6s** | **65.8s** | **89.7s** | **104.1s** | **109.3s** |
| GNS | 200 | 7056 points | 43.5s | 73.5s | 111.6s | 162s | 226.2s | 305.9s | 386.0s |
| GAT | 200 | 7056 points | 146.5s | 236.5s | 394.2s | 532.8s | 645.2s | 733.8s | 812.5s |

Table 8: Contrasting the change in computation time with an increase in graph size at a fixed radius of 0.060. The times shown represent the overall time needed to infer all 200 time steps. We compare our time-stepper against other neural network based physics solvers

| MODEL | PARAMETERS | CONNECTIVITY | MATERIAL | TIME STEPS | GRAPH SIZE | | | |
|---|---|---|---|---|---|---|---|---|
| | | | | | 1776 POINTS | 4143 POINTS | 5608 POINTS | 7056 POINTS |
| OURS | 4,312,247 | 0.060 | Plasticine | 200 | **3.9s** | **14.5s** | **27.3s** | **47.6s** |
| GNS | 1,592,987 | 0.060 | Plasticine | 200 | 7.8 s | 38.3s | 68.7s | 111.6s |
| GAT | 1,999,003 | 0.060 | Plasticine | 200 | 71.1s | 153.4s | 295.3s | 394.2s |

Table 9: Contrasting the inference times (in seconds) for highly dense point clouds up-sampled from highly sparse graphs (1776 points).

| TIME STEPS | ROLLOUT SIZE | ROLLOUT TIME (s) | FULL-ORDER SIZE | UPSCALE TIME (s) |
|---|---|---|---|---|
| 200 | 1776 | 3.9 | 7,000 | 5e-5 |
| 200 | 1776 | 3.9 | 40,000 | 3e-4 |
| 200 | 1776 | 3.9 | 60,000 | 8e-4 |
| 200 | 1776 | 3.9 | 100,000 | 9e-4 |

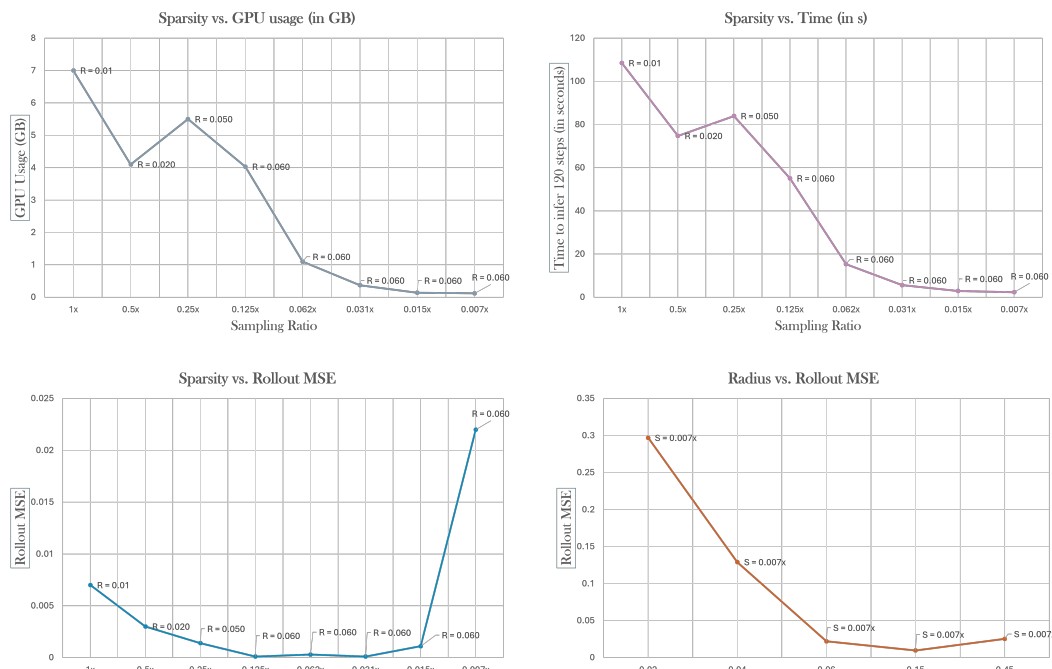

Figure 6: **Effects of sparsity on Rollout Loss on the Elasticity dataset (78k)** The above graphs highlight how the time-stepper performance at different sparsity settings (as a ratio of 78k). The graph of sparsity vs. GPU usage highlights the highest GPU usage at the specified radius of the input graph. The Sparsity vs. Time graph highlights the computation time as a function of sparsity, at the specified input graph radius. The Sparsity vs. Rollout MSE graph shows that the at 0.125x, 0.062x and 0.031x, the model achieves a rollout loss of the order of 1e-4. To show that increasing the radius doesn't always improve performance, we show in the bottom right graph that on the sparsest graph (0.007x), the MSE increases when the radius is increased beyond 0.06.

**Number of message-passing layers**  We show that the key bottleneck in terms of speed is the message-passing operation within the interaction network. This operation scales with the number of edges as $E = O(K^2)$, where $K$ is the number of nodes.

Table 10: This table shows that the number of message-passing layers results in a negligible improvement in rollout Loss.

| NUM. MESSAGE PASSING LAYERS | CONNECTIVITY | INPUT SIZE | INFERENCE TIME/STEP | LOSS |
|---|---|---|---|---|
| 2 | 0.077 | 2247 | 3.6 | 0.0008 |
| 4 | 0.077 | 2247 | 3.8 | 0.0009 |
| 6 | 0.077 | 2247 | 4.2 | 0.0014 |
| 8 | 0.077 | 2247 | 4.3 | 0.0009 |

**Graph radius and viscosity**  We observed that during inference, larger neighborhoods resulted in greater rigidity within the system. The following table highlights the changes in viscosity as the

graph radius varies during inference. We measure viscosity by the highest average velocity attained by the particles and the lowest average distance between them.

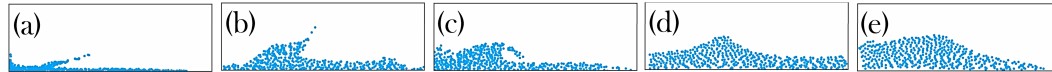

Figure 7: The figure highlights the increase in viscosity as the radius increases, due to a larger neighborhood, **(a)** with a radius of 0.010, **(b)** 0.015, **(c)** 0.025, **(d)** 0.040, **(e)** 0.055.

Table 11: This table highlights the increase in viscosity, measured by the average minimum inter-particle distance over 200 time steps and the average maximum velocity over 200 time steps. Higher values of minimum inter-particle distance denote a more rigid graph where the particles don't collide with each other as often, and a lower average particle velocity indicates a more constrained flow

| RADIUS | AVG. MIN INTER-PARTICLE DIST. | AVG. MAX PARTICLE VELOCITY |
|---|---|---|
| 0.015 | 6.06e-5 | 1.7e-2 |
| 0.020 | 6.32e-5 | 8.1e-3 |
| 0.025 | 6.4e-5 | 7.6e-3 |
| 0.030 | 6.42e-5 | 6.6e-3 |
| 0.035 | 6.51e-5 | 5.7e-3 |
| 0.040 | 6.58e-5 | 5.0e-3 |
| 0.045 | 6.71e-5 | 4.6e-3 |
| 0.050 | 7.04e-5 | 4.2e-3 |

**Discretization invariance w.r.t the latent grid**   We show that the model is agnostic to the resolution of the latent grid during inference. If the latent grid is too small, the performance degrades due to data loss. However, with larger latent grid sizes, there is no significant improvement in performance. The model was tested on latent grid dimensions of 8, 16, 32, 64, and 128. The results are shown in Figure 9.

**Effects of latent grid**   The latent grid allows the Fourier neural operator to learn the temporal dynamics on a regular grid of fixed size. This allows it to learn the dynamics of non-uniform and complex geometries. Table 12 shows the performance of the model in the absence of the latent grid.

Table 12: The table showcases invariance to grid sizes greater than 8. At sizes less than 16, the model fails to perform as well due to data loss

| GRID SIZE | ROLLOUT LOSS |
|---|---|
| 128 | 0.0072 |
| 64 | 0.0074 |
| 32 | 0.0081 |
| 16 | 0.0075 |
| 8 | 0.0110 |

## G.1   SAMPLING STRATEGY VS. ROLLOUT LOSS

We compared different sampling strategies against the rollout Loss (MSE). The results are presented in Table 13.

Table 13: Comparison of different sampling and graph construction strategies against Rollout MSE on Water-2D dataset

| SAMPLING STRATEGY | GRAPH TYPE | ROLLOUT MSE |
|---|---|---|
| RANDOM | RADIUS | 0.0098 |
| RANDOM | DELAUNAY | 7.017 |
| FPS | RADIUS | 0.0097 |
| FPS | DELAUNAY | 8.04 |

## G.2 EFFECTS OF NOISE ON ROLLOUT ACCURACY

Temporal auto-regressive models suffer from corruption of simulations due to noise accumulation. The attention mechanism used in this architecture helps mitigate this issue to an extent. However, when the system is too chaotic, such as 3D water simulations, it is important to choose the right noise scale to ensure that the model is robust to this noise accumulation. We experimented with different noise standard deviations and found that values between 1e-4 and 3e-4 resulted in the most stable rollouts for 3D Water simulations. This can be observed in Figure 8.

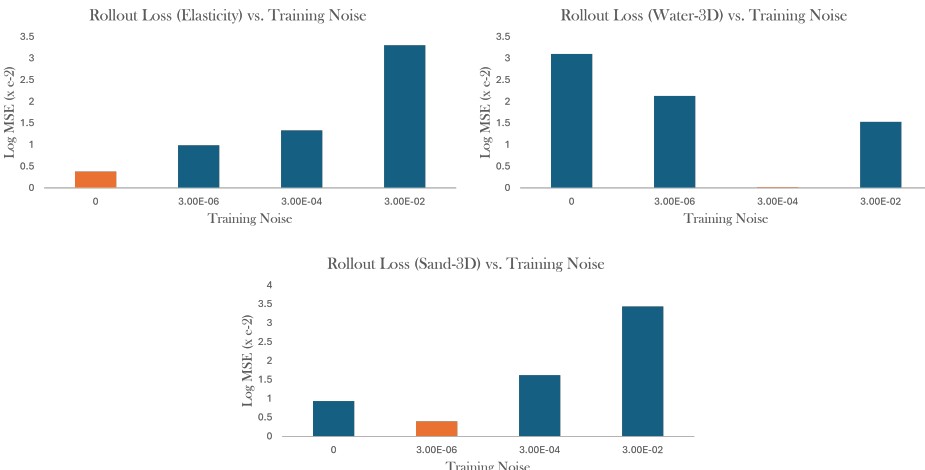

Figure 8: **Effects of noise** on different physical systems

**Effect of training dataset size on generalizability** We performed experiments to see if the performance would improve significantly with the addition of new trajectories in the `WaterDrop2D` dataset. We observed that the rollout loss steadily decreases with the addition of new trajectories. However, this decrease is less apparent after 200 trajectories. The loss is much higher when the number of trajectories is less than 100.

Table 14: This table shows the trends in rollout loss with the number of training trajectories for the Water-2D dataset. The model generalizes fairly well when trained on 150 trajectories, after which there's a gradual improvement in performance

| TRAINING SIZE (# TRAJS) | ROLLOUT LOSS (MSE) |
|---|---|
| 50 | 0.010 |
| 150 | 0.0067 |
| 200 | 0.0064 |
| 400 | 0.0061 |
| 1000 | 0.0059 |

## G.3 DESIGN DECISIONS WITH MINIMAL IMPACT

We performed hyper-parameter tuning on `WaterDrop` dataset and found that the following parameters have the least impact on the overall model performance. The results are illustrated in Figure 9.

**Time window for input velocity sequence** The window used for input velocity sequence didn't affect the accuracy of the output by a significant amount. We experimented with window sizes of [2, 3, 5, 6, 7]. A window size of 2 allows for the network to be a Markov process, with the model predicting the acceleration at a time step from the corresponding acceleration at the previous time step. This can be leveraged for interactive manipulation of the material within the simulation.

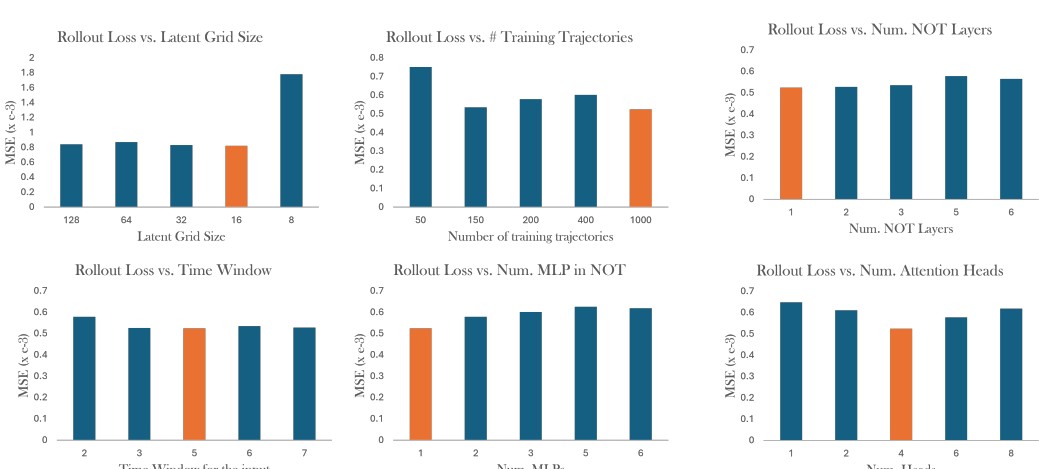

Figure 9: **Hyperparameters with minimal impact** The above graphs show minimal effects of some of the hyperparameters on the rollout loss on the WaterDrop dataset.

**Number of MLP layers in NOT**    We experimented with [1, 2, 4, 8, 10] as the number of layers within the architecture. The accuracy decreased slightly with more MLPs.

**Graph reduction and discretization invariance**    We approximately account for the integration weights of the GNO in equation 7 by computing the mean of the kernel values in each neighborhood. Note that we would not obtain a neural operator when using a sum as a reduction method since the values would diverge in the limit of finer discretizations.

**GNO hidden layer size**    We tried three configurations of GNO hidden layers for both the non-linear kernel and the linear kernel, i.e, [32, 64], [512, 256], and [64, 512, 1024, 256]. In the latter cases, the models became significantly bulkier without a noticeable change in performance.

**Number of transformer layers**    We experimented with [1, 2, 4] layers of the transformer and found that a single layer outperformed the rest. As the architecture became bulkier, the model's tendency to converge at local minima slightly increased.

**Number of attention heads**    The performance of the model improved with more attention heads, however, it became a little worse after increasing the number of heads beyond 4.

**Number of experts**    The model performed optimally with 2 experts and the performance slightly degraded for all other settings.

**GNO radius**    We experimented with the following radii - [0.0004, 0.0015, 0.015, 0.045, 0.100]. In each of these cases, there wasn't a noticeable change in the performance or the inference times. x

**Projection layers**    We experimented with three configurations for projection layers, i.e., [1, 2, 5], and observed only minor variations in the performance. To optimize for the parameter count, we chose a single projection layer in the final model.

