# OpenReview forum: "Reduced-Order Neural Operators: Learning Lagrangian Dynamics on Highly Sparse Graphs"
_ICLR.cc/2025/Conference — Submitted to ICLR 2025_

### Official Review · Reviewer_eAsE · 2024-10-26

**Soundness:** 3
**Presentation:** 3
**Contribution:** 3
**Rating:** 6
**Confidence:** 3

**Summary:**

The article introduces Graph Informed Optimized Reduced-Order Modeling (GIOROM), a framework designed to accelerate simulations of Lagrangian dynamics—such as fluid flows, granular flows, and elastoplasticity—using neural-operator-based reduced-order modeling. GIOROM addresses this by simulating physics on highly sparse graphs sampled from the spatially discretized system, achieving a reduction in input size by 6.6 to 32 times compared to full-order models.

To capture local spatial features of the discretized input, the authors define a graph-based neural operator called the Interaction Operator, which performs two key tasks:

1. It uses a discretization-agnostic adaptation of message passing to model interactions between points, regardless of the discretization.

2. It leverages a Graph Neural Operator (GNO) layer to project features onto a regular grid, facilitating the construction of a latent space upon which the time-stepper operates.

Authors use following scheme: Discretization-agnostic MP ->  GNO ->  point-wise MLP -> Neural operator transformer -> GNO -> discretization-agnostic MP. This scheme predicts acceleration field in Q-point discretization which is used for computing velocity in t_(n+1) and deformation in t_(n+1) (using explicit Euler scheme) and after that combined loss is calculated between ground truth derformations and velocities (in this discretization). Neural field is trained using reconstruction loss and used for full-order inference (to P-point discretization).Developed approach generalizes well across different initial conditions, velocities, discretizations and geometries.

**Strengths:**

1. Developed approach generalizes well across different initial conditions, velocities, discretizations and geometries.

2.  The paper includes a thorough ablation study, which evaluates the impact of different components of the model. This study helps in understanding how each part contributes to the overall performance, thereby validating the effectiveness of the proposed methods.

3.The authors provide strong scientific justifications for their approach, supported by rigorous experiments.

**Weaknesses:**

1. Only explicit time integrator is considered. Please add motivation for this choice to paper.
2. Sometimes units in tables are not presented.

**Questions:**

1. Please clarify units of duration for Table 1.
2. Please add information about numerical scheme used in nclaw. Is it the same scheme used for training model?
3. What about stability issues of numerical scheme used? How the time step is chosen?
4. How does the developed approach compare with implicit numerical schemes in terms of stability and time-stepping requirements?

---

> ### Author Response · Authors · 2024-11-22
>
> Q1. Only explicit time integrator is considered. Please add motivation for this choice to paper.
>
> A1. Indeed, the present work only considers explicit time integration. This is consistent with prior machine–learning–based time stepping methods. (See Learning to Simulate Complex Physics with Graph Networks). We agree that it is exciting to work on alternative time integration schemes, such as Runge-Kutta and implicit Euler, in the future.
>
> Q2. Please clarify units of duration for Table 1.
>
> A2. We have clarified these in the updated PDF. The time interval is 5e-3s per time-step.
>
> Q3. Please add information about numerical scheme used in nclaw. Is it the same scheme used for training model?
>
> A3. Yes, nclaw also uses an explicit time integration scheme. That said, the training data for the elastic examples (owl-shaped mesh from the LiCROM paper) uses an implicit integration scheme. Since our model works on both of these settings, we show that, our work is compatible with training data of any time integration scheme.
>
> Q4. What about stability issues of numerical scheme used? How the time step is chosen?
>
> A4. Consistent with other machine-learning-based methods (e.g., GNS, CROM), our approach remains stable even if we take a larger time step than the training data generated using classical numerical methods. However, as the reviewer pointed out, our approach also becomes unstable if too big of a time step is chosen. All our examples are generated with the same time step as the training data, whether they are generated using explicit (sand, nclaw dataset) or implicit scheme (elasticity, LiCROM dataset). We will further clarify in this paper and add a stability analysis (w.r.t. the time step size).
>
> Q5. How does the developed approach compare with implicit numerical schemes in terms of stability and time-stepping requirements?
>
> A5. As discussed in the previous answer, our approach can take the same time step size as the training data generated using implicit numerical schemes (elasticity, LiCROM dataset). It will be interesting for future work to explore the possibility of integrating implicit time integration with machine-learning-based force evaluation, which, to our knowledge, has not been done before. This will require differentiating the neural operator and graph neural network during runtime.

---

> > ### Comment · Reviewer_eAsE · 2024-11-25
> > **Thank you**
> >
> > Thank you for your answers. I raise my score to 6.

---

### Official Review · Reviewer_oDsv · 2024-10-29

**Soundness:** 2
**Presentation:** 1
**Contribution:** 2
**Rating:** 3
**Confidence:** 3

**Summary:**

The paper introduces Graph Informed Optimized Reduced-Order Modeling, a neural-operator-based framework designed to accelerate simulations of complex Lagrangian dynamics, such as e.g. fluid flows. GIOROM intends to reduce computational costs by training on sparse graphs sampled from spatially discretized systems, allowing it to simulate temporal dynamics efficiently without depending on full spatial resolution.
The authors emphasize that the proposed framework is discretization-invariant, meaning it can generalize across various spatial discretizations and resolutions.  To achieve this, the framework uses a graph-based neural operator transformer to model temporal dynamics on sparse representations and leverages continuous reduced-order modeling with neural fields to reconstruct the full-order solution, enabling evaluation at any spatial point within the system.

**Strengths:**

This approach shows some innovation by integrating graph-based neural networks and continuous reduced-order techniques, distinguishing it from more conventional dense simulations and the results point out computational gains. This Acceleration while retaining high accuracy can benefit industries where large-scale physics simulations are essential, e.g. engineering.

**Weaknesses:**

The paper's novelty is either limited or not effectively communicated, as the work primarily appears to combine approaches from the cited prior studies [Li et al., 2024; Chen, 2024]. The rationale behind the specific design choices and the reasons for the approach's effectiveness are not clearly explained.

Significant revision is needed, particularly in Sections 3, 4 and 5.  Sections 3 and 4 introduce the methods used but fail to clarify how the proposed work differs from existing research or to explain how the components fit together cohesively. The introduction is challenging to follow, especially since it doesn’t clarify how this work advances beyond prior literature. A more logical approach could be to introduce the complete framework as depicted in Figure 1 and follow the flow of this diagram, highlighting any novel contributions.

While the experimental nature of the work in Section 5 and in the Appendix is valuable, the structure of the experimental section lacks a clear line. It consists of nine figures and tables with minimal explanation. To improve clarity, the authors should provide a more detailed rationale for each experiment, explaining the intentions behind the choices and the methodologies used. Tables and figures might benefit from rearrangement or relocation to the appendix if necessary.

Although the paper claims high fidelity across various initial conditions and resolutions, it does not address complex cases with extreme variability, such as materials undergoing phase transitions or highly turbulent flows. These more complex scenarios, which are generally more challenging to model with reduced-order methods, are not covered in the results presented.

The research in this work is quite interesting; however, I cannot recommend the paper for acceptance in its current form as major revision is needed.

**Questions:**

How does the model handle edge cases with high variability or extreme dynamics? It would strengthen the work to clarify how robust the model is for complex, highly dynamic systems, such as those with rapid phase changes or intense turbulence.

What are the limitations of sparse sampling in terms of capturing fine-grained details? Can you provide information of the trade-offs between computational efficiency and the accuracy of capturing  system details?

How does the model perform when interpolating or extrapolating to new materials or conditions not represented in the training data? It is mentioned that these cases are challenging but no empirical evidence is provided.

Can you provide a systematic study on the scaling performance with varying graph size for the same dynamical system?

Are there specific physics-based scenarios where this reduced-order approach might be less effective?

---

> ### Author Response · Authors · 2024-11-22
>
> **Major Revisions**
>
> We have revised the paper as suggested. However, we welcome any further feedback on any parts of the paper that require improvements in presentation.
>
> 1. The paper's novelty is either limited or not effectively communicated
>     A. We have re-written the introduction section to better clarify our novelty and distinguish our method from prior approaches.
> 2. Sections 3, 4 - explanations of how they fit together, Section 5 - Additional explanation for experiments.
>     A. We provided explanations at the start of sections 3 and 4, explaining how they relate to Figure 1. The introductory section further distinguishes how the parts explained in sections 3 and 4 differ from prior literature. We furthermore added additional details about the experiments in section 5.
>
> 3. To improve clarity, the authors should provide a more detailed rationale for each experiment, explaining the intentions behind the choices and the methodologies
>     A. Each experiment has been updated with a paragraph explaining the experimental setup, evaluation metric, and the reasoning for the experiment.
>
> Q1.  Although the paper claims high fidelity across various initial conditions and resolutions, it does not address complex cases with extreme variability, such as materials undergoing phase transitions or highly turbulent flows. These more complex scenarios, which are generally more challenging to model with reduced-order methods, are not covered in the results presented.
>
> A1. Indeed, our current approach can have issues with discontinuities and stress concentration (e.g., shear localization). We will add this to the discussion/limitation section of our paper. Increasing the sampling resolution can partially alleviate the issue, but tradeoffs between accuracy and the number of samples have to be made (See Figure 6). Importance sampling methods that increase sampling density in the singular geometry region can lead to more efficient results (See Contact-centric deformation learning, ACM SIGGRAPH 2022; Optimizing Cubature for Efficient Integration of Subspace Deformations, ACM SIGGRAPH 2008). The SPH-based Newtonian fluid datasets used in the paper exhibit chaotic behavior. We will release the dataset containing these trajectories, similar to what is presented in the GNS paper [Sanchez-Gonzales, 2020]. Modeling phase transitions is a consideration for future work which will solidify reduced-order modeling
>
> Q2. What are the limitations of sparse sampling in terms of capturing fine-grained details? Can you provide information of the trade-offs between computational efficiency and the accuracy of capturing system details?
>
> A2. We have updated Figure 6 in the new Pdf to contrast accuracy, GPU usage, computation-time against sparsity and graph radius. We observe a decrease from ~7GB to <1GB on 0.031x sampled graphs, while achieving a rollout MSE of the order of 1e-4.
>
> Q3. How does the model perform when interpolating or extrapolating to new materials or conditions not represented in the training data? It is mentioned that these cases are challenging but no empirical evidence is provided.
>
> A3. We used a large number of trajectories and showed that the model can handle unseen trajectories with unseen conditions, generated by random initial velocity vector generated by random seeds. We also showed that the model can handle multi-material scenarios shown in Table 1. However, the model is unable to generalize to unseen materials because each material is encoded within the encoder and for unseen material, the encoding would be undefined or would be approximated to the nearest known material.
>
> Q4. Can you provide a systematic study on the scaling performance with varying graph size for the same dynamical system?
>
> A4. Yes, this has been provided in the updated figure 6. We observed lower GPU usage, faster computation time and comparable performance with higher levels of sparsity, upto 0.031x the full-order point cloud (78k particles). Increasing sparsity beyond 0.031x leads to degradation of performance.
>
> Q5. Are there specific physics-based scenarios where this reduced-order approach might be less effective?
>
> A5. When the system is not bounded and the number of particles in the system is not fixed, the reduced-order representations may fail to capture newer information that was previously not in the system. An example could be a fluid flow system with particles constantly entering and leaving the system of vortices.

---

### Official Review · Reviewer_LDdW · 2024-11-01

**Soundness:** 3
**Presentation:** 3
**Contribution:** 2
**Rating:** 5
**Confidence:** 5

**Summary:**

This paper develops a technique for simulating physical systems by combining graph-informed ROM and neural operators. First, the initial full-order mesh is coarse-grained to a sparser mesh using the farthest point sampling method. That smaller graph is encoded via an interaction network by performing several message passing, capturing local spatial interactions. Then, the encoded features are embedded into a regular grid, processed with a neural operator transformer and decoded with a second interaction network in order to perform the integration step. Each snapshot computed on the reduced space can be projected back to a full-order system by using a learnt linear basis transformation, which can be efficiently solved with least squares and evaluated at any arbitrary point. The method is applied to several examples in both solid and fluid mechanics.

**Strengths:**

* The method is very versatile, as it is able to handle different types of systems and material models in continuum mechanics problems. It has also very good results in generalization.

* The processing pipeline is discretization-agnostic, and the solution can be sampled in an arbitrary level of discretization.

* Solving the dynamics in the reduced space makes the algorithm very fast in inference time.

**Weaknesses:**

* The graph construction is computationally demanding.

* The method could be problematic with challenging geometries or singular phenomena, in which the farthest point sampling might omit relevant details of the domain.

* The pipeline is too complicated and has many hyperparameters, which might be a problem for learning larger-scale systems and more complicated physical behaviours.

**Questions:**

* It is very surprising that the method is very complicated on the encoding stage (going from full order to reduced order) but fairly simple on the decoder stage. In my opinion, the paper has not complitely justified why the use of a graph encoder + interaction network is better than other existing reduction methods. I would suggest the authors to perform an ablation study comparing the current graph encoder approach with a neural field encoding, so there is clear evidence that the use of a sparsified graph and an interaction network is better than a more simple approach.

* Could the authors to provide a sensitivity analysis showing how performance and computational cost vary with different values of Q (number of reduced point discretization)? Which are the heuristics or guidelines used for selecting Q in practice? This would help readers understand the tradeoffs involved in selecting this parameter.

* Line 164: "This ensures an even distribution of points, [...]". This is an advantage from the computational perspective, but it might be problematic under certain conditions. For example, pressure discontinuities in fluid dynamics or stress concentration in solid mechanics. Those singular regions might not be correctly captured by just using a distance criterion in the farthest point algorithm, and might be advantegous a finer discretization in those areas. Can the authors provide with any discussion/limitations on this regard?

* Figure 2: The bunny and cow examples show self-contact between surfaces. Is there any specific condition to handle this situations from the graph generation perspective, and is it handled differently in solids and fluids? Could the authors discuss any limitations on this area?

* Figure 3: The figure is misleading. Why is the farthest point graph more dense than the full-order graph? I would expect something like the Delaunay Graph example.

* Table 3: Can the authors clarify what "randomizing the indices" mean and its significance in the reduction method?
My guess is that the authors refer to different ordering in snapshots and nodal indices. If that is the case, it is not obvious why that randomization should affect the final result in a relevant way. In fact, POD/PCA decomposition is constructed over the snapshot covariance matrix and it is invariant under any permutation transformation.

* Table 3: Can the authors provide more details about the autoencoder baseline and the LiCROM projection, including parameter counts, training times and network topology? Could the authors include a more fair comparison with a closer method, such as its non-linear version (CROM)?

Final comment: This paper has very promising results and generalization capabilities, but the contributions are not well justified. The real contribution of the paper is the graph sparse reduction together with an interaction operator, as the use of neural operator transformers and LiCROM neural fields is not novel. In the current state of the paper, there is a lack of comparison with other state-of-the-art end-to-end reduction techniques. For this reason, my final rating is marginally below the acceptance threshold, but I would be open to raising it with enough justification of my mentioned points.

**Details Of Ethics Concerns:**

I have no ethics concerns.

---

> ### Author Response · Authors · 2024-11-22
>
> Q1. The graph construction is computationally demanding.
>
> A1. We show in Table 4 that using Graph layers allows us to achieve the best performance since our inputs are irregular and neural operators such as FNO, GNOT, DINO etc. are designed to work on regular grids. GINO does not account for inter-particle interactions, which is crucial for Lagrangian Dynamics. In the absence of which, the models fail to capture spatial information, leading to suboptimal performance. We created computation benchmarks in Figure 6 of the updated PDF which shows how graph size affects the GPU usage and computation time as a function of graph radius and sampling percentage.
>
> Q2. Handling discontinuities/limitations with farthest point sampling
>
> A2. Indeed, our current approach can have issues with discontinuities and stress concentration (e.g., shear localization). Increasing the sampling resolution can partially alleviate the issue, but tradeoffs between accuracy and the number of samples have to be made (See Figure 6 ). Importance sampling methods that increase sampling density in the singular geometry region can lead to more efficient results (See Contact-centric deformation learning, ACM SIGGRAPH 2022; Optimizing Cubature for Efficient Integration of Subspace Deformations, ACM SIGGRAPH 2008).
>
> Q3. Complicated Pipeline, too many hyper-parameters.
>
> A3. We agree with the reviewers that this model has several hyperparameters. However, we have shown in the config files that, once tuned, many of these hyperparameters can be considered universal hyperparameters, that work across physical systems, with standard deviation of the noise being one of the few features that changes across them. We require only little system-specific tuning. Also, its flexibility and modularity can also be seen as an advantage of being able to tailor it to more complicated systems (informed choices of subsampling, neighborhood selection, edge information, smoothness prior in the decoder).
>
> Q4. Complexity of Encoder (with interaction operator) , Decoder
>
> A4. We would like to clarify that the order reduction is done through farthest point sampling and not using learnable techniques. The encoder is used to capture the local spatial interactions. This has been clarified in the updated Figure 1. However, the use of Neural Field as the encoder would speed up the process by removing the message passing. As such we performed an ablation study with NF-NOT-NF setup on the elasticity dataset, where the order-reduction is learnable through the NF encoder. However, we observed that NF was not as efficient as the Interaction Operator in capturing spatial dependencies and interactions. We present the results below:
>
> NF-NOT-NF: 1-step MSE: 1.26e-6, RolloutMSE: 1.35
> IO-NOT-IO: 1-step MSE: 5.07e-10, RolloutMSE: 4e-4
>
> Q5. Handling self-contact
>
> A5. The training data is generated by the material point method (MPM), which does not explicitly check collision and implicitly handles self-collision through a background grid, for both solids and fluids. We follow a similar approach as MPM as well as the GNS baseline by letting the neural network to implicitly these self contacts. This has been added to the discussion section. Better fine-grained self-contact sampling is an exciting future work direction (See Contact-centric deformation learning, ACM SIGGRAPH 2022). We will add it to the future work.
>
> Q6. Figure 3: The figure is misleading. Why is the farthest point graph more dense than the full-order graph?
>
> A6. We have updated Figure 3. to include N(odes), E(dges), R(adius) for each of the graphs. FPS based graph has more edges because it uses a larger radius than the Full-order system. The edges connect points that are farther away from each other. Delaunay graph on the other hand has a fixed number of edges, thus making it discretization-dependent. Having edges connecting points further away allows us to efficiently model long-range relationships, which due to the properties of graph layers, is retained if the density increases. However, importantly, if the density decreases, adding new edges by increasing the radius of the input during inference, allows the model to still infer the correct dynamics. This has been shown in Figure 6 (sparsity vs. accuracy). Furthermore, to justify that the speedup is not affected due to the addition of new edges, we show in Tables 9 and 10 that our setup outperforms other time-steppers in terms of speed even with more edges.

---

> > ### Author Response · Authors · 2024-11-22
> >
> > Q7. Effect of randomization
> >
> > A7. Thank you for the clarifying question. We have clarified this further in the updated paper. In a nutshell, we do not mean changing the order in snapshots or nodal indices. Rather, we meant that after training, the POD/PCA/autoencoder methods could not evaluate information other than the original mesh on which they were trained. To verify this, we pass in different meshes of the domain (created by randomizing the indices) and confirm that they indeed do not work. In fact, any mesh that is not the original trained mesh would not work with POD. Unlike POD, the neural field-based approach is discretization invariant and can evaluate arbitrary mesh whether they are created by indices randomization or not. We will clarify this in the paper and add additional experiments where the alternative mesh is created through other techniques, for example, different triangulation of the domain.
> >
> > Q8. Details about the autoencoder baseline and the LiCROM projection. ?
> >
> > A8. The details regarding the autoencoder baseline are been provided in the updated PDF. As per the suggestion of Reviewer WgNe, we have removed Table 3, and replaced it with numbers within the discussion. Our method is compatible with CROM as well, i.e., replacing the LiCROM components with CROM. Unfortunately, CROM entails higher computation costs (equation 4 becomes nonlinear least squares and thereby require highly expensive solvers) and struggle with handling highly nonlinear deformations (see comparison between CROM and LiCROM in figure 14 of https://arxiv.org/pdf/2310.15907).
> >
> > Q9. Sensitivity Analysis on Q
> >
> > We have updated Figure 6 with different analyses on the effect of sampling.

---

> > > ### Comment · Reviewer_LDdW · 2024-11-25
> > >
> > > I thank the authors for the assesment and aclaration of my previous comments. I would like to make some comments about the rebuttal.
> > >
> > > Q6: I thank the authors for the updated Figure. Now it is more clear that the farthest point graph has less nodes than the full order graph.
> > >
> > > Q7: Thank you for the clarification, now the explanation makes more sense.
> > >
> > > Q4: This was my major concern about the paper, and in my opinion it has not been addressed correctly. I understand the motivation of using the IO to account for local interactions, but the paper still lacks of clear evidence that the "reduction + encoding" method is better than any other recent discretization-invariant MOR technique. I would have liked to see in all the examples of the paper, in a similar fashion to Table 5, the performance of GIOROM with respect to LiCROM kind of reduction to clearly show that the sparse graph + IO step is totally necessary for the accuracy of the method.
> > >
> > > I think that the paper is in better shape after the rebuttal and thank the authors for the changes, but based on the comments above I would like to keep my initial score.

---

> > > > ### Author Response · Authors · 2024-12-02
> > > >
> > > > Q4. We agree with you that the paper needs a table comparing against baseline MOR architectures. We are currently running experiments to provide the numbers. We apologize that due to the time-constraints, we are unable to provide an entire table as of now, but we do have some results that we would like to share with you.
> > > >
> > > > Our work fundamentally differs from recent discretization-invariant MOR (Model-Order Reduction) techniques (e.g., LiCROM, CROM) in the sense that our work is a “non-intrusive” MOR method while those methods are “intrusive” methods. This means that even after they are trained, LiCROM and CROM require the PDE solver to time-step during inference.
> > > >
> > > > Intrusive methods like LiCROM and CROM require that both the underlying equations are known, including the detailed material parameters. Without explicit knowledge about the equation, these approaches cannot do time integration. Moreover, they also require that the corresponding numerical method code (e.g., FEM, MPM, SPH) used to generate the data is available. Such a requirement prevents them from applying to real-world engineering problems where the underlying equation or the traditional numerical simulation code is unavailable.
> > > >
> > > > By contrast, GIOROM is a “non-intrusive” MOR technique thanks to our neural-operator-based time integration module (i.e., the interaction operator IO + Neural Operator Transformer). With that, we do not require any explicit knowledge of the underlying equation or any material parameters, across the entire pipeline.
> > > >
> > > > To validate this, we performed the experiment where we used the pre-trained LiCROM setup to infer the plasticine system using just the data and no explicit information about the PDE and we show that LiCROM fails to capture the system as effectively, in the absence of Interaction Operator, which can generalize to data and does not require any explicit PDE knowledge. We present the result on the nclaw plasticine dataset:
> > > >
> > > > GIOROM
> > > > roll out 0.00014372089775667015
> > > >
> > > > LiCROM without knowledge of physics
> > > > roll out 0.01592236121586893
> > > >
> > > > We will provide the results on the other systems in the final version of the pdf and apologize for not providing them here due to time constraints.
> > > >
> > > > LiCROM (During Inference)
> > > > Known PDE -> [Order Reduction] -> FEM/MPM time-stepping with PDE knowledge -> [Increase to FOM with neural field]
> > > >
> > > > GIOROM (During inference)
> > > > Unknown PDE -> [Order Reduction] -> Neural Operator + data input -> [Increase to FOM with neural field]
> > > >
> > > > We provide an anonymized link to some visualizations we have generated to showcase that in the absence of PDE information, LiCROM time-stepping fails
> > > >
> > > > https://www.dropbox.com/scl/fo/l1kf7le8px45bqbnuyb77/AIa61zqE5A-1iaXh1LUrS6M?rlkey=d5qaf0alar86752n4o67i5gfb&st=o8xk7v2y&dl=0

---

### Official Review · Reviewer_WgNe · 2024-11-04

**Soundness:** 3
**Presentation:** 3
**Contribution:** 3
**Rating:** 6
**Confidence:** 4

**Summary:**

The paper proposes a neural operator approach to Lagrangian simulations. The core idea is to subsample the original point cloud, then transform it into/from a fixed-sized latent (grid) representation with Message Passing (MP) + Graph Neural Operator (GNO), and in the middle apply a transformer. This model is used to predict the acceleration at each node, which is then numerically integrated to evolve the system. While aligned with Graph Neural Networks (GNNs) for Lagrangian dynamics, the proposed method is discretization-independent and consistently speeds up simulations compared to GNNs.

**Strengths:**

- Extensive literature review.
- Detailed analysis of graph subsampling and construction implementations.
- Many details and ablations of the proposed approach, including an extensive appendix.

**Weaknesses:**

- **W1: self-generated datasets**: Is there any reason why you didn't use existing 3D datasets, e.g., from Sanches-Gonzalez et al. (2020)? The only subset that is not there is a 3D multi-material one. Looking at the baseline results in your Table 4, GNS and the proposed model perform similarly. Comparability with the baseline numbers from the GNS paper would have been very useful.
- **W2: Farthest Point Sampling (FPS)**: You say on lines 351-352 that random sampling has a similar performance to FPS, but on L. 329, you say that you do use FPS after all. Having some experience with FPS, I know that it is a sequential process over the point cloud, with the number of iterations being the number of subsampled points. And this sequential nature becomes a bottleneck when working with large point clouds as the presented 3D ones. Why didn't you just use random sampling?
- **W3: POD and Autoencoders**: What do you mean by POD and an autoencoder in Table 3? Do you explain this somewhere? I don't think any of these two are representative baselines. In addition, Table 3 has multiple identical lines, all of which could be summarized in one sentence. Please consider removing this table altogether.
- **W4: DINo**: You cite the DINo paper (Yin et al., 2023) and a few similar ones, but I didn't see a discussion of how your approach improves on them. To my knowledge, you could have used the DINo encoder and decoder on the velocity field (and Euler-integrate once), and you would only have had to add an operator transformer in between. Am I missing something?

**Minor:**
- L. 310, "Ma et al. (2023)" -> "(Ma et al., 2023)"
- L. 412: please add some space between the captions of subplots (a) and (b). Currently, it is unclear that there are two subplots.
- Table 13: please reduce the font size of this table
- L. 1048: "our model is at least 5x faster than [GNNs]" doesn't match with the numbers in Tables 9 and 10. It is rather a 2-4x speedup. Please reformulate.

**Questions:**

- Which dataset do you use for Table 3?
- Table 3: Did you consider discussing "ROM baselines" (L.431) and "Discretization-agnostic ROM" (L.514) next to each other, so that Table 3 is close to both?

---

> ### Author Response · Authors · 2024-11-22
>
> Thank you for your feedback
>
> Q1. Is there any reason why you didn't use existing 3D datasets, e.g., from Sanches-Gonzalez et al. (2020)? The only subset that is not there is a 3D multi-material one. Looking at the baseline results in your Table 4, GNS and the proposed model perform similarly. Comparability with the baseline numbers from the GNS paper would have been very useful.
>
> A1. The NCLAW datasets were generated with complex geometries, allowing us to test discretization invariance and geometry invariance. 3D GNS datasets offer low flexibility for testing these invariance features since their 3D datasets do not have complex geometries.
>
> Q2. You say on lines 351-352 that random sampling has a similar performance to FPS, but on L. 329, you say that you do use FPS after all. Having some experience with FPS, I know that it is a sequential process over the point cloud, with the number of iterations being the number of subsampled points. And this sequential nature becomes a bottleneck when working with large point clouds as the presented 3D ones. Why didn't you just use random sampling?
>
> A2. We show in Table 13 that the two sampling strategies have comparable performance. While our point-clouds were small enough (<100k points), we used FPS for even distribution of points, on larger point-clouds, random sampling may be used without loss of accuracy.
>
> Q3. What do you mean by POD and an autoencoder in Table 3? Do you explain this somewhere? I don't think any of these two are representative baselines. In addition, Table 3 has multiple identical lines, all of which could be summarized in one sentence. Please consider removing this table altogether.
>
> A3. For reduced-order methods (ROM), POD and autoencoders are standard baselines. See [Model reduction of dynamical systems on nonlinear manifolds using deep convolutional autoencoders] by Lee et al. We have added more explantions and background citations for these two methods. We apologize for the identical lines. We have removed the table from the paper.
>
> Q4. Which dataset do you use for Table 3?
>
> A4. We have updated the paragraph to include the dataset used for computing the metrics. We used The Owl Dataset (Elasticity) for all the experiments shown in (now removed) Table 3.
>
> Q5. Table 3: Did you consider discussing "ROM baselines" (L.431) and "Discretization-agnostic ROM" (L.514) next to each other, so that Table 3 is close to both?
>
> A5. Thank you for the recommendation. We have updated it in the revised pdf.
>
> Q6. You cite the DINo paper (Yin et al., 2023) and a few similar ones, but I didn't see a discussion of how your approach improves on them. To my knowledge, you could have used the DINo encoder and decoder on the velocity field (and Euler-integrate once), and you would only have had to add an operator transformer in between. Am I missing something?
>
> A6. DINo works on grid based inputs. Therefore, it is necessary to include integral transform operation to convert from unstructured pointcloud to a structured grid, before passing it to DINo. We performed experiments with DINo as the processor on the elasticity dataset, in place of NOT, and observed similar performance as NOT, showing that it is a valid time-stepper.
>
> IO-DINo-IO: 1-step MSE: 6.25e-10, Rollout MSE: 7e-4
>
>
> Q7.
>
> The following changes have been made in the updated pdf.
>
>
> L. 310, "Ma et al. (2023)" -> "(Ma et al., 2023)"
> L. 412: please add some space between the captions of subplots (a) and (b). Currently, it is unclear that there are two subplots.
> Table 13: please reduce the font size of this table
> L. 1048: "our model is at least 5x faster than [GNNs]" doesn't match with the numbers in Tables 9 and 10. It is rather a 2-4x speedup. Please reformulate.</b>

---

> > ### Comment · Reviewer_WgNe · 2024-11-26
> >
> > Q1: I disagree that your datasets are harder to learn because your initial shapes are actually not so relevant. This is just the initial state of the point cloud; from there on, it is just a material falling on the floor, very much the same as in GNS.
> >
> > Q2/3/4/5/6/7: Thanks.
> >
> > Overall, I agree with reviewer oDsv that your contribution is rather limited - solving the GNS task with a combination of GNO (encoder/decoder) + Transformer (processor) is nothing groundbreaking. However, the sheer amount of ablations (e.g., graph subsampling and construction) is a reason to keep my score at 6 "above the acceptance threshold."

---

> > > ### Author Response · Authors · 2024-11-26
> > >
> > > We appreciate your feedback and thank you for providing constructive criticism that helped us improve the quality of the paper.
> > >
> > > Q1. We agree with your assessment that our datasets are not harder than GNS dataset. We apologize for prior miscommunication that our dataset is more difficult to learn. In fact, "This is just the initial state of the point cloud; from there on, it is just a material falling on the floor, very much the same as in GNS", this is precisely what we wanted to verify when we chose different geometries. We used similar setup as GNS dataset for water and sand, choosing the same style of trajectories, duration, with similar boundary conditions, with only difference being the shape of the object because we wanted to confirm the geometry invariance of our model.

---

### Official Review · Reviewer_iRQR · 2024-11-04

**Soundness:** 3
**Presentation:** 2
**Contribution:** 2
**Rating:** 6
**Confidence:** 3

**Summary:**

The paper proposes a learning-based reduced-order modeling framework for Lagrangian simulation. The proposed model comprises  message passing layer and operator transformer.  To reduce the computational complexity, the original discretized field is first down-sampled and then reconstructed via linear combination of neural field basis. Extensive numerical experiments on different Lagrangian simulation including fluids and sand are conducted to demonstrate the effectiveness of the proposed method.

**Strengths:**

1. Combining a continuous decoder with a neural operator in the latent space is technically sound. The numerical experiments showcase the proposed framework significantly reduces the computational cost while maintaining good accuracy. The continuous decoding strategy also makes it more flexible with different discretizations.

**Weaknesses:**

1. Many details about the experiments are vaguely described, which makes it difficult to interpret some of the results presented.
 For example, in equation 4, how is the coefficient q derived in practice? Is it predicted by another neural network, or is it optimized directly via least-squares? If it is optimized online, then during the inference stage, it will require extra optimization for every reconstruction from reduced mesh to full mesh.  In table 4, several baselines are listed but there is no formal definition or a brief introduction of them, for example I could not find the definition of what is IPOT. In table 3, what is the autoencoder? Is it a GNN or it's just a MLP? If it's a GNN then shouldn't it be permutation-equivariant?

2. As shown in Figure 1, the reduction part does not contain any learnable parts but rather just rely on non-learnable sampling method, which can potentially result in information loss.

3. (Relatively minor) The system considered in the numerical experiments are rather small-scale, all less than 100k nodes/particles.

**Questions:**

1. When applying FNO as part of the latent dynamics model, how do you handle the empty voxels/regions?

2. The paper compares  with full-order model like GNS or other reduce-order model, how does the model compare to multi-grid model like Cao et al. [1]?

3. Is it necessary to do the sampling of full-order mesh at every timestep, and why not stay in the latent space with a fixed set of particles sampled in the beginning?


[1] Cao Y, Chai M, Li M, et al. Efficient Learning of Mesh-Based Physical Simulation with BSMS-GNN[J]. arXiv preprint arXiv:2210.02573, 2022.

---

> ### Author Response · Authors · 2024-11-22
>
> Thank you for your feedback.
>
> Q1. In equation 4, how is the coefficient q derived in practice? Is it predicted by another neural network, or is it optimized directly via least-squares? If it is optimized online, then during the inference stage, it will require extra optimization for every reconstruction from reduced mesh to full mesh.
>
> A1. Yes, it is optimized directly via a linear least squares. We follow equation 10 from https://arxiv.org/pdf/2310.15907. This boils down to solving a symmetric positive linear system using a single Cholesky factorization. Therefore there is no computationally expensive optimization involved (faster than a neural network evaluation), thereby introducing minimum overhead (See Table 9 upscale time, ~1e-4s). We have added this in the paper.
>
> Q2. In table 4, several baselines are listed but there is no formal definition or a brief introduction of them, for example I could not find the definition of what is IPOT.
>
> A2. We apologize for this oversight. These models have been defined in the updated Pdf, along with the relevant citations.
> GNN - Graph Neural Network, GAT - Graph Attention Network, GINO - Geometry Informed Neural Operator, GNOT - General Neural Operator Transformer, IPOT - Inducing Point Operator Transformer.
>
> Q3. In table 3, what is the autoencoder? Is it a GNN or it's just a MLP? If it's a GNN then shouldn't it be permutation-equivariant?
>
> A3. It is an MLP autoencoder, similar to the ones commonly used in standard reduced-order models. See [Model reduction of dynamical systems on nonlinear manifolds using deep convolutional autoencoders] by Lee et al. Our approach, on the other hand, guarantees permutation-equivariance and allows for evaluation at arbitrary locations in space. This has been clarified in the updated Pdf.
>
> Q4. As shown in Figure 1, the reduction part does not contain any learnable parts but rather just rely on non-learnable sampling method, which can potentially result in information loss.
>
> A4. Yes, information loss is possible. As we discussed in Figure 6 (left), there is a trade-off between accuracy and the number of points sampled. Indeed, it is an exciting future work to explore learnable methods to potentially reduce information loss. Furthermore, the “potential information loss” during training enhances the time-stepper model robustness and accelerates training. For inference, we can take any resolution—in particular, also omit subsampling.
>
> Q5. When applying FNO as part of the latent dynamics model, how do you handle the empty voxels/regions?
>
> A5. The FNO is applied to functions on an equidistant grid. The integral transform before and after the FNO allows us to change the discretization from a general point cloud, as done in GINO [Li et al., 2024]. In particular, function values on the latent grid originate from learned aggregations over neighborhoods with a fixed radius, i.e., having more neighbors if the particles are denser.
>
> Q6. The paper compares with full-order model like GNS or other reduce-order model, how does the model compare to multi-grid model like Cao et al. [1]?
>
>
> A6. The Interaction Operator presented in the paper can be converted to a multi-level graph model, which, as shown in "Multipole Graph Neural Operator for Parametric Partial Differential Equations, Li et al.", indeed behaves as a neural operator. It is an exciting future research direction to see how the improved time complexity of multi-grid operators can make learning Lagrangian dynamics faster and more efficient.
>
> Q7. Is it necessary to do the sampling of full-order mesh at every timestep, and why not stay in the latent space with a fixed set of particles sampled in the beginning?
>
> A7. During training, to increase robustness, the inputs are partitioned into slices of 6 time-steps, 5 for input and 1 for output. All 6 frames have the same set of particles. However, other slices may have a different set of particles. During inference, the model predicts a fixed set of particles sampled at the initial time-step over the entire temporal sequence. An additional feature supported by the model, is that the second (decoder) integral transform layer can infer particles that were not in the input graph. This theoretically allows us to have varying set of particles (between input and output) at each time-step. This feature was, however not tested because we saw no application of it in the scenarios shown in the paper.

---

### Author Response · Authors · 2024-11-22

We sincerely thank the reviewers for the valuable feedback they provided, which has helped us improve the clarity of presentation.

We have made major revisions to the paper as suggested by Reviewer oDsv, with the revised sections highlighted in red in the updated PDF.
These include but are not limited to
1. Clearly stating the contribution and how it differs from prior work in the introduction
2. Providing better explanations at the start of sections 3 and 4, regarding how they relate to Figure 1 and the over all ROM setup.
3. Explanation of motivation behind each experiment conducted in Section 5
4. Additional discussions that were suggested by the reviewers.

Common Responses:
Q1. Justification of the architecture and how it differs from prior work.
A1. We present a new mathematical formulation - The Interaction Operator, which takes into account inter-particle interactions, used in Lagrangian Dynamics. Prior implementation of Integral Transform [GINO, Li et al, 2024], was designed to work on Eulerian formulations with no temporal dynamics and thus did not take into account local particle interactions.

Q2. Vagueness of Experimental Setup
A2. This has been addressed in the revised pdf, with explanations provided for all the experiments in section 5 and their corresponding tables.

---

### Meta-Review · Area_Chair_jYoa · 2024-12-20

**Metareview:**

The submission deals with Lagrangian simulation and proposes a method based on discretization, downsampling and reconstruction. Five reviewers generally appreciated the paper but raised several weaknesses:

- Novelty,
- Lack of details, clarity,
- the limiting role of farthest point sampling,
- complexity of the method and justification of the contributions,
- Small scale experiments, simplicity of the problems,
- No standard datasets used,
- Positioning,

The authors could provide answers for some of these issues, but some important problems remained. The AC sides with the critical reviews and judges that the paper is promising but not yet ready for publication. The decision is mainly based on several critical issues: the refusal to use standard datasets and lacking of justifications of the necessity of sparsification.

**Additional Comments On Reviewer Discussion:**

The reviewers engaged with the authors, and discussed the paper with the AC.

---

### Decision · Program_Chairs · 2025-01-22

Reject